# Fourier-transform-based attribution priors improve the interpretability and stability of deep learning models for genomics

**Alex M. Tseng**[1]    **Avanti Shrikumar**[1]    **Anshul Kundaje**[1,2]

[1]Department of Computer Science, [2]Department of Genetics
Stanford University
{amtseng, avanti, akundaje}@stanford.edu

## Abstract

Deep learning models can accurately map genomic DNA sequences to associated functional molecular readouts such as protein–DNA binding data. Base-resolution importance (i.e. "attribution") scores inferred from these models can highlight predictive sequence motifs and syntax. Unfortunately, these models are prone to overfitting and are sensitive to random initializations, often resulting in noisy and irreproducible attributions that obfuscate underlying motifs. To address these shortcomings, we propose a novel attribution prior, where the Fourier transform of input-level attribution scores are computed at training-time, and high-frequency components of the Fourier spectrum are penalized. We evaluate different model architectures with and without our attribution prior, training on genome-wide binary labels or continuous molecular profiles. We show that our attribution prior significantly improves models' stability, interpretability, and performance on held-out data, especially when training data is severely limited. Our attribution prior also allows models to identify biologically meaningful sequence motifs more sensitively and precisely within individual regulatory elements. The prior is agnostic to the model architecture or predicted experimental assay, yet provides similar gains across all experiments. This work represents an important advancement in improving the reliability of deep learning models for deciphering the regulatory code of the genome.

## 1 Introduction

Transcription factors (TFs) are proteins that regulate gene activity with extraordinary specificity by recognizing and binding to short DNA sequence patterns—or "motifs"—in regulatory elements of the genome. High-throughput experiments have been used to profile genome-wide regulatory activity in diverse cell types and tissues [1]. For example, chromatin immunoprecipitation sequencing experiments (e.g. ChIP-seq, ChIP-exo, and ChIP-nexus) provide binding readouts of specific targeted TFs, and chromatin accessibility experiments (e.g. DNase-seq and ATAC-seq) provide aggregate readouts of all protein–DNA contacts, for each base (DNA letter) in the genome [1]. Deep neural networks—particularly convolutional neural networks (CNNs)—have achieved state-of-the-art performance in mapping DNA sequence to TF binding and chromatin accessibility profiles [2, 3, 4]. These models accept fixed-length DNA sequence segments of the genome as an input, and—through a series of convolutional layers—predict molecular labels as measured by a regulatory profiling experiment. A common goal of these genomic deep learning models is to ultimately identify the regulatory sequence code (i.e. motifs and their syntax) underpinning genomic regulation, and this has motivated the development of a wide suite of tools to infer base-pair-resolution importance (i.e. "attribution") scores in the input DNA sequences to reveal the regulatory code [4, 5, 6, 7].

Unfortunately, the high capacity of these models makes them prone to overfitting on training-set noise rather than true signal [8]. Querying these deep learning models for their attribution scores often reveals their reliance on regions of the input sequence that are not biologically relevant to the task at hand. Perhaps worse, their attributions are irreproducible and highly sensitive to random initializations (Figure 1). This poses a hindrance to the reliable identification of motifs driving genomic regulation from the attribution scores.

There is a need to improve the interpretation of high-capacity genomic deep learning models with the goal of downstream motif discovery. To address this need, we propose a novel attribution prior [9, 10] based on Fourier transforms to directly reward the notion of interpretability during model training. That is, at training time, we impose a secondary loss function that penalizes the network for improper attributions, thereby *explicitly* training the model to maximize interpretability along with correctness of predictions. We show that our Fourier-based attribution prior can be flexibly applied to different model architectures using a diverse set of real experimental data, and offers improvements in interpretability, motif discovery, and learning stability (Figure 1).

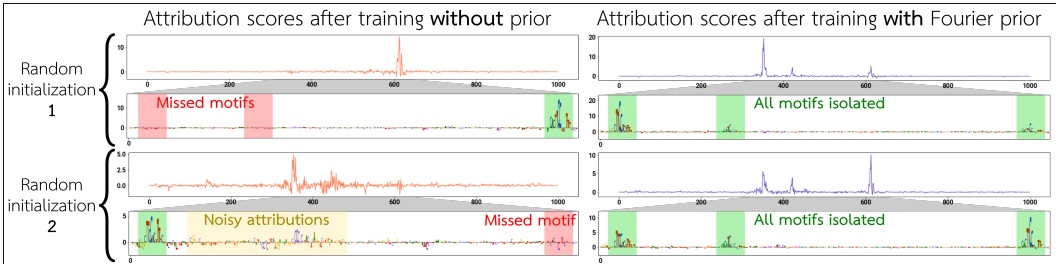

Figure 1: Models trained with the standard approach (left) irreproducibly miss motifs in the underlying sequence and noisily rely on irrelevant regions of the input. When training with the Fourier-based attribution prior (right), models consistently and cleanly identify the driving motifs. The examples shown are from binary binding models of the SPI1 TF from TF ChIP-seq experiments [1]. See Supplementary Figures S14–S15 for more examples.

## 2 An attribution prior for genomics based on Fourier transforms

### 2.1 Formulation of the Fourier-based attribution prior

Regulatory DNA sequences are sparsely composed of short functional sequence motifs ($\sim$6–20 bases in length) with soft syntactic constraints on motif combinations, density, spacing, and orientation. To maximize interpretability with the goal of motif recovery, CNNs trained on regulatory data would ideally place importance (i.e. attribution) only on informative motifs, and little importance on the irrelevant background sequence. Let $x$ represent a one-hot encoded input sequence to the model, $y$ represent the true labels, and $f$ represent the model prediction function. Let $g(x, f)$ represent the per-position attribution scores on the input $x$, where $g(x, f)$ is a vector of attributions of length equal to that of $x$. We train our models with a two-part loss function: $L(x, y, f) = L_c(f(x), y) + \lambda L_p(g(x, f))$, where $L_c$ is the standard correctness loss for the model, and $L_p$ is the attribution prior loss. The choice of $g$ depends on the model type (Supplementary Methods Sec. 2.3). In this work, we formulate $g$ using input gradients for computational efficiency, and subsequently show that the improvements granted by the prior follow through to an alternative reference-based attribution method.

Our Fourier-based prior loss is computed by taking the Fourier transform of the attribution vector $g(x, f)$. Let $m$ denote the magnitudes of the positive-frequency Fourier components of (a slightly smoothed version of) $g(x, f)$, and $\hat{m}_i$ denote the $i^{\text{th}}$ component of $\frac{m}{\|m\|_1}$ (smaller $i$ correspond to lower frequencies) (Supplementary Figure S1).

We penalize the high-frequency components in $\hat{m}$ as follows:

$$L_p(g(x, f)) = 1 - \sum_{i=1}^{L} w_i \hat{m}_i$$

Where

$$w_i = \begin{cases} 1 & i \leq T \\ \frac{1}{1+(i-T)^s} & i > T \end{cases}$$

This attribution prior penalizes the model for attempting to place importance along the input DNA sequence in bursts shorter than the limit set by $T$, which denotes the *minimum* length of sequence that can be considered a reasonable motif. This limit is softened by the parameter $s$, which controls the rate at which this penalty is smoothly reduced for higher frequencies. $T$ is set based on prior knowledge by assuming no motif will be shorter than 6–7 base pairs (bp), while $s$ is fixed to $0.2$ for all models (Supplementary Methods Sec. 2.3). For other training tasks where the expected motif length is different, $T$ and $s$ may be adjusted accordingly. Note that due to the $\ell_1$-normalization of $\hat{m}$, the value of $L_p$ is bounded between 0 and 1. Fourier transforms are linear and very fast to calculate, resulting in an attribution prior $L_p$ that is efficient to compute and differentiate. Furthermore, by using Fourier transforms, our prior is effectively invariant to the locations and the number of motifs within an input sequence.

## 2.2 Training data and model architectures

To demonstrate the flexibility of the Fourier-based prior, we train on four different experimental datasets over two different model architectures. Our experimental datasets target the SPI1 protein (TF ChIP-seq) in 4 cell types [1]; the GATA2 protein (TF ChIP-seq) in 3 cell types [1]; the Nanog, Oct4, and Sox2 proteins (TF ChIP-seq) in mouse embryonic stem cells [4]; and chromatin accessibility (DNase-seq) in the K562 leukemia cell line [1]. These datasets vary not only in the type of experimental assay being predicted, but also in the number of motifs and the complexity of motif syntax driving the measurements. For each dataset, we train both binary models and profile models. Binary models predict a binary label for each 1000-base-pair-long input sequence (i.e. whether or not there is a statistically significant peak in the TF ChIP-seq or DNase-seq accessibility signal overlapping the center of the sequence [1]) (Supplementary Figure S2). Profile models, on the other hand, predict base-resolution TF ChIP-seq or DNase-seq profiles from each 1346 bp input sequence, and therefore are able to finely track motifs based on associated patterns in the shape of the output profile (Supplementary Figure S3) [4].

When we train binary models with the Fourier-based prior, the prior loss weight $\lambda$ is set to 1. On profile models, the prior loss weight is selected to be half of the value of the correctness loss on a model trained without the prior. In both architectures, this ensures that the value of the prior loss is on a similar scale as the correctness loss (as recommended by Ross et al. [10]—see Supplementary Methods Sec. 2.3 and Supplementary Figure S4).

## 3 Fourier-based priors improve signal-to-noise ratio for the detection of predictive motifs

We consider models trained with and without the Fourier-based prior, and compare the interpretability of the models using DeepSHAP [11] scores, which extend DeepLIFT [7] backpropagation-based attributions using Shapley values. While the attribution prior computed at training time operated on input gradients due to efficiency, we rely on DeepSHAP attribution scores to assess a model's interpretability because DeepSHAP scores provide base-pair-level attributions against a biologically meaningful background—in this case, shuffled versions of the input sequence that preserve dinucleotide frequencies (as recommended by Shrikumar et al. [7]).

Over all of our datasets and model architectures, we find that training with the Fourier-based prior significantly denoises attribution scores and dramatically improves the detection of predictive motifs near ChIP-seq/DNase-seq peak summits (local maxima in the profiles around which driver motifs are expected to be found). A model that elevates attributions only at the motifs—and has flat attributions everywhere else—is expected to have a reduction in the high-frequency Fourier components and the Shannon entropy of the input sequence attributions (Supplementary Methods Sec. 4). We see a reduction in both metrics on all of our datasets and architectures (Supplementary Table S2, Supplementary Figure S5). Using a Wilcoxon rank test, all improvements in test-set interpretability when training with the Fourier-based prior are significant at the $1 \times 10^{-6}$ level.

We visually show the improvement in interpretability over examples of peak sequences in the test set, including the ability of models trained with the Fourier-based prior to cleanly highlight motifs (Figure 2, Supplementary Figure S6).

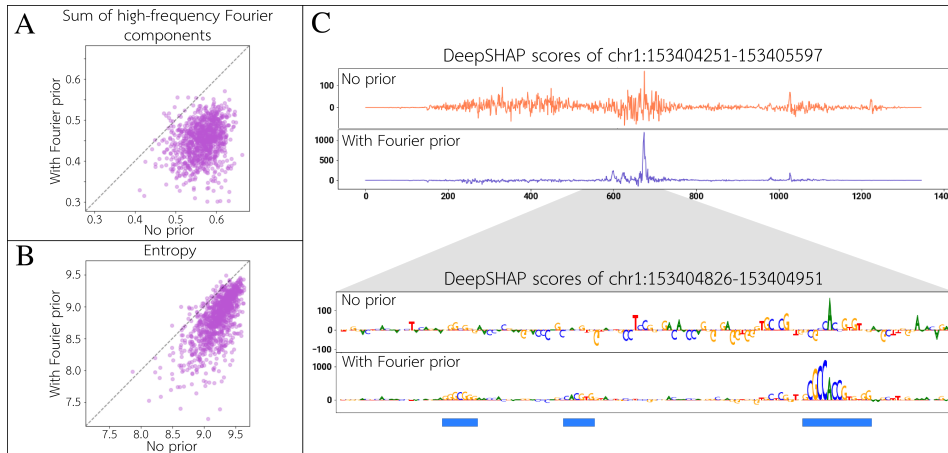

Figure 2: On K562 DNase-seq profile models, for each sampled test-set sequence, we compare the: **A)** sum of high-frequency normalized Fourier components; and **B)** Shannon entropy of the DeepSHAP attributions between models trained with versus without the Fourier-based prior. **C)** At a particular K562 open chromatin peak, we show the attributions across the entire input sequence, and the base-pair-level attributions around the summit region. The model trained with the Fourier-based prior cleanly highlights 3 motifs centered around the peak summit, matching relevant transcription factors (left to right: SP1, CLOCK, and CTCF).

## 4 Fourier-based priors improve sensitivity and specificity of motifs

On multi-task models trained to predict binding of 3 TFs (Nanog, Oct4, and Sox2), we perform motif discovery and motif instance calling, and compare the calls to independently collected gold-standard ChIP-nexus experiments [4] that can highlight putative bound motifs at very high ($\sim$10 bp) resolution. We use TF-MoDISco, a motif discovery approach, to distill recurring TF-binding motifs across multiple input sequences from base-resolution importance scores [6]. TF-MoDISco first finds predictive subsequences of high importance (called "seqlets") across all input sequences. Seqlets are subsequently aligned, clustered, and summarized into a non-redundant set of motif patterns. We then use these motifs to scan test-set peak sequences and their attributions to call high confidence matches to the motif (i.e. motif instances). For both model architectures, we find that the Fourier-based prior improves the quality of motifs discovered by TF-MoDISco, as well as the sensitivity and specificity of called motif instances supported by ChIP-nexus peaks (Supplementary Figures S7–S9).

We show an example of the improvement in motif discovery and motif calls using the multi-task Nanog/Oct4/Sox2 profile models. Specifically, on the Nanog prediction task trained with the Fourier-based prior, TF-MoDISco identified the known ATCAA and GGAAAT Nanog motifs (among others), in addition to the Oct4-Sox2 motif [4]; the Nanog motifs were missed without the prior (Figure 3). Additionally, called motif instances originating from the model trained with the prior show substantially improved support from independent high-resolution ChIP-nexus data.

## 5 Fourier-based priors improve prediction performance of binary models

Whereas profile models learn to predict continuous ChIP-seq or DNase-seq read coverage profiles at base resolution, the more popular models trained on lower-resolution binary labels (e.g. bound vs. unbound) do not benefit from profile shape information, and thus are particularly prone to overfitting [4]. On all of our binary models, we see an improvement in validation- and test-set performance when training with the Fourier-based prior (Supplementary Figure S11). Table 1 shows the best test-set performance achieved when training with versus without the Fourier-based prior, over 30 random initializations each.

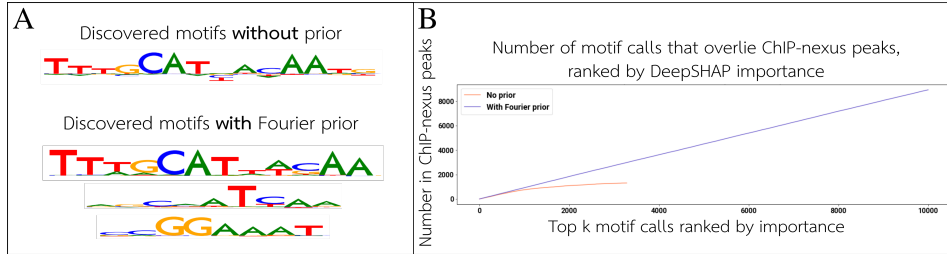

Figure 3: **A)** We show a subset of relevant motifs for the Nanog binding prediction task discovered by TF-MoDISco when training with versus without the Fourier-based prior. The ATCAA and GGAAAT Nanog motifs [4] are missed by the model trained without the prior. **B)** Using the motifs discovered by TF-MoDISco, we call motif instances on test-set peak sequences. Ranked by total attribution magnitude, we compute a cumulative count of motif calls that overlap gold-standard bound motif instances from independent high-resolution Nanog ChIP-nexus data. The prior substantially improves the recall of these gold-standard motif instances.

We also show a more detailed depiction of the performance distribution for binary models trained to predict SPI1 binding (Figure 4).

Training on only 1% of the training data reveals a similar trend, showing that the Fourier-based prior assists in improving model generalizability even when the model is trained on sparse data (Supplementary Figure S12).

Table 1: Binary model performance on test set

|  | Test accuracy | | Test auROC | | Test auPRC | |
| --- | --- | --- | --- | --- | --- | --- |
|  | No prior | With prior | No prior | With prior | No prior | With prior |
| SPI1 | 87.996% | **88.347%** | 0.954 | **0.956** | 0.853 | **0.858** |
| GATA2 | 85.519% | **86.216%** | 0.924 | **0.928** | 0.733 | **0.745** |
| Nanog/Oct4/Sox2 | 84.640% | **85.437%** | 0.906 | **0.908** | 0.655 | **0.661** |
| K562 | 90.049% | **90.362%** | 0.966 | **0.968** | 0.963 | **0.966** |

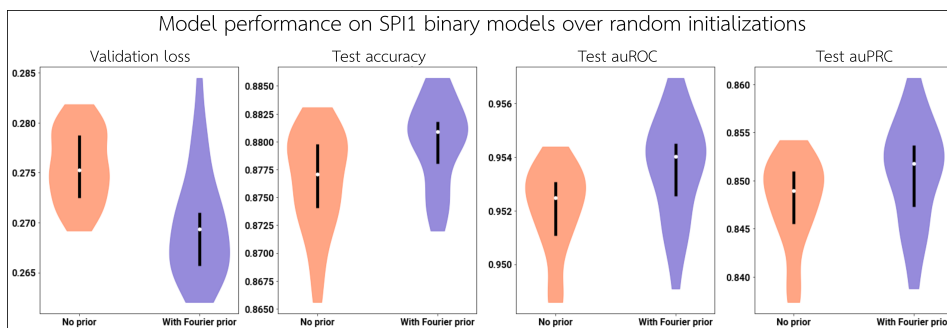

Figure 4: We train binary models to predict SPI1 binding over 30 random initializations without the prior, and another 30 with the Fourier-based prior. We show the distribution of the validation loss, test-set accuracy, test-set auROC, and test-set auPRC across the 30 random initializations in each condition.

## 6 Fourier-based priors improve stability of attribution scores

Over all of our datasets and model architectures, we find a dramatic improvement in the stability of attributions when models are trained with the Fourier-based prior. For each dataset, on a sample of test-set peak sequences, we compute the similarity of DeepSHAP attribution scores between the models to quantify how consistently the models learned on each particular sequence. We evaluate

consistency between two sets of attribution scores using continuous Jaccard similarity—a metric used by TF-MoDISco—as it is designed for comparing similarity between two importance score tracks, accounting for both sequence and attribution scores [6] (Supplementary Methods Sec. 7).

The Fourier-based prior improves the stability of the attribution scores across random initializations, thereby allowing robust inference of motifs (Table 2; Supplementary Figures S13–S15). The improvements in learning stability are even stronger between models trained on significantly smaller training sets (i.e. 1% of the original set) versus the entire training set (Table 2; Supplementary Figures S13, S16). This suggests that using the prior, models trained on sparse data learn more similarly to when they are trained with the full dataset. Using a Wilcoxon rank test, all improvements in test-set stability when training with the Fourier-based prior are significant at the $1 \times 10^{-6}$ level.

Table 2: Learning stability on test-set sequences (continuous Jaccard similarity)

| | | Similarity across random initializations | | Similarity between training on all vs 1% data | |
| --- | --- | --- | --- | --- | --- |
| | | No prior | With prior | No prior | With prior |
| Binary | SPI1 | $95.6 \pm 1.0$ | $\mathbf{134.3 \pm 1.0}$ | $29.8 \pm 0.7$ | $\mathbf{42.0 \pm 0.8}$ |
| | GATA2 | $105.8 \pm 1.5$ | $\mathbf{140.5 \pm 1.8}$ | $29.6 \pm 1.3$ | $\mathbf{43.5 \pm 1.4}$ |
| | Nanog/Oct4/Sox2 | $107.0 \pm 2.5$ | $\mathbf{129.0 \pm 2.2}$ | $43.1 \pm 1.3$ | $\mathbf{53.1 \pm 1.7}$ |
| | K562 | $121.1 \pm 2.0$ | $\mathbf{197.5 \pm 2.5}$ | $48.3 \pm 1.7$ | $\mathbf{67.2 \pm 1.8}$ |
| Profile | SPI1 | $332.9 \pm 11.2$ | $\mathbf{405.6 \pm 11.4}$ | $77.6 \pm 4.4$ | $\mathbf{233.1 \pm 9.8}$ |
| | GATA2 | $369.1 \pm 10.1$ | $\mathbf{390.5 \pm 9.5}$ | $102.9 \pm 5.2$ | $\mathbf{121.6 \pm 5.9}$ |
| | Nanog/Oct4/Sox2 | $216.7 \pm 5.7$ | $\mathbf{246.5 \pm 4.8}$ | $66.9 \pm 4.5$ | $\mathbf{109.7 \pm 6.2}$ |
| | K562 | $210.2 \pm 5.8$ | $\mathbf{236.0 \pm 4.2}$ | $31.4 \pm 1.5$ | $\mathbf{121.3 \pm 5.2}$ |

# 7 Fourier-based priors increase specificity of model reliance on regions of regulatory function

When we train our models with the Fourier-based prior, we generally observe that the models are able to place higher attribution in high-confidence regions of regulatory significance, and less attribution on other irrelevant areas (Table 3). To show this, we consider test-set peak sequences, and compute the Spearman correlation between each base pair's DeepSHAP importance to its distance from the peak summit. A more negative correlation implies that the model is placing higher importance closer to the most confident region of binding, as determined by the experimental assay. Additionally, we compute a rank-based measure of specificity by calculating the precision and recall of individual bases in the input sequences that overlap the precise ∼200 bp peak regions when ranked by importance (Supplementary Figure S17). A higher area under the precision–recall curve implies that the model places higher importance in called peaks, and in a more specific/precise manner. Using a Wilcoxon rank test, all improvements in summit distance correlation when training with the Fourier-based prior are significant at the $1 \times 10^{-6}$ level.

Table 3: Placement of importance on summit/peak regions in test set

| | | Correlation of importance to distance from summit | | auPRC of overlap with called peaks | |
| --- | --- | --- | --- | --- | --- |
| | | No prior | With prior | No prior | With prior |
| Binary | SPI1 | $-0.433 \pm 0.002$ | $\mathbf{-0.442 \pm 0.002}$ | $0.595$ | $\mathbf{0.616}$ |
| | GATA2 | $-0.392 \pm 0.003$ | $\mathbf{-0.419 \pm 0.003}$ | $0.608$ | $\mathbf{0.617}$ |
| | Nanog/Oct4/Sox2 | $-0.509 \pm 0.003$ | $\mathbf{-0.544 \pm 0.003}$ | $\mathbf{0.674}$ | $0.669$ |
| | K562 | $-0.335 \pm 0.005$ | $\mathbf{-0.360 \pm 0.005}$ | $0.587$ | $\mathbf{0.594}$ |
| Profile | SPI1 | $-0.464 \pm 0.005$ | $\mathbf{-0.490 \pm 0.004}$ | $0.398$ | $\mathbf{0.470}$ |
| | GATA2 | $-0.391 \pm 0.006$ | $\mathbf{-0.437 \pm 0.005}$ | $0.430$ | $\mathbf{0.497}$ |
| | Nanog/Oct4/Sox2 | $-0.454 \pm 0.005$ | $\mathbf{-0.476 \pm 0.005}$ | $0.464$ | $\mathbf{0.516}$ |
| | K562 | $-0.392 \pm 0.006$ | $\mathbf{-0.411 \pm 0.006}$ | $0.530$ | $\mathbf{0.548}$ |

For the Nanog/Oct4/Sox2 TF-binding and the K562 chromatin accessibility datasets, we use additional independent binding data to further assess the ability of a model to place importance in biologically relevant regions. For Nanog/Oct4/Sox2 binding, we consider high-resolution binding peaks called from independently collected ChIP-nexus experiments [4], and for K562 chromatin accessibility, we consider a set of high-resolution binding footprints explicitly derived from several DNase-seq profiles using independent signal-processing methods [12]. These ChIP-nexus peaks and footprints define high-confidence, high-resolution regions where bound regulatory motifs are expected to be found. For these datasets, we further demonstrate that models trained with the Fourier-based prior are able to place importance more specifically in relevant regions, by quantifying the fraction of attributions that overlie a ChIP-nexus peak or DNase-seq footprint, and by considering the precision–recall of important regions overlapping with the ChIP-nexus peaks or DNase-seq footprints (Table 4, Supplementary Figures S18–S19). Using a Wilcoxon rank test, all improvements in importance overlap fraction with ChIP-nexus peaks or DNase-seq footprints when training with the Fourier-based prior are significant at the $1 \times 10^{-6}$ level.

Table 4: Importance placement on independently derived ChIP-nexus peaks or DNase-seq footprints

|  |  | Fraction of importance in ChIP-nexus peak/footprint | | auPRC of overlap with ChIP-nexus peaks/footprints | |
| --- | --- | --- | --- | --- | --- |
|  |  | No prior | With prior | No prior | With prior |
| Binary | Nanog/Oct4/Sox2 | $0.806 \pm 0.006$ | $\mathbf{0.830 \pm 0.006}$ | 0.644 | **0.647** |
|  | K562 | $0.272 \pm 0.005$ | $\mathbf{0.281 \pm 0.005}$ | **0.235** | 0.229 |
| Profile | Nanog/Oct4/Sox2 | $0.689 \pm 0.006$ | $\mathbf{0.787 \pm 0.006}$ | 0.598 | **0.630** |
|  | K562 | $0.236 \pm 0.004$ | $\mathbf{0.298 \pm 0.004}$ | 0.215 | **0.256** |

The Fourier-based prior consistently improves interpretability in all experiments shown, with only one exception: the precision–recall of importance overlapping peaks/footprints, and *only* with *binary* models trained to predict Nanog/Oct4/Sox2 binding or K562 chromatin accessibility (on our profile models, the prior always improves the auPRC). We emphasize that this is not a failure of the prior, but a symptom intrinsic to the binary architecture. These two datasets comprise the more complex tasks in this study: in addition to the primary motifs, several secondary motifs can indirectly contribute to binding or accessibility, even though they do not directly underlie a peak [13, 14]. *Profile* models are able to finely track motifs along shifting peaks, allowing them to isolate specifically the primary motifs. *Binary* models, however, see the same sequences repeatedly to predict a binary label, so for these complex tasks with an abundance of secondary motifs, the model would also rely on motifs that do not underlie peaks/footprints. The Fourier-based prior improves the identification of *all* relevant motifs and increases their relative importance. Thus, on a binary architecture, the prior highlights motifs outside of ChIP-nexus peaks or DNase-seq footprints, because *they are still informative for a binary label*. In concordance with this observation, we find that for these complex binary tasks, the regions outside ChIP-nexus peaks or K562 footprints that are highlighted by the Fourier-based prior contain motifs of important secondary TFs (Supplementary Figure S20) [4]. On the intrinsically limited binary architecture, learning to rely on these secondary motifs is the correct thing to do, even though it directly reduces the computed precision of importance overlap in peaks/footprints for these complex tasks.

Finally, we examine models trained on simulated sequences and evaluate the ability of the Fourier-based prior to focus importance on specific individual motifs. On simple binary models trained to predict binding of the SPI1 TF from synthetic sequences, training with the Fourier-based prior resulted in a significantly higher fraction of importance being placed in motif instances on average $(0.659 \pm 0.027$ vs. $0.218 \pm 0.010, p < 1 \times 10^{-6}$ by Wilcoxon test). Additionally, models trained with the Fourier-based prior had a much higher auPRC of importance-ranked base overlap with motif instances (0.607 vs. 0.422) (Supplementary Figure S22).

# 8 Unlike existing regularization techniques, Fourier-based priors simultaneously optimize performance and interpretability

## 8.1 Fourier-based prior versus L2-regularization

We compare SPI1 binary models trained with L2-regularization (i.e. weight decay) versus the Fourier-based prior. Although L2-regularization was able to improve the predictive performance of the model more than the improvement brought on by the Fourier-based prior, L2-regularization also severely hurts model interpretability. We find that DeepSHAP attributions from the model trained with the Fourier-based prior have a significantly lower sum of high-frequency Fourier components ($0.374 \pm 0.002$ vs. $0.400 \pm 0.002$, $p < 1 \times 10^{-6}$ by Wilcoxon test) and a significantly lower Shannon entropy ($7.891 \pm 0.011$ vs. $8.908 \pm 0.014$, $p < 1 \times 10^{-6}$ by Wilcoxon test) (Supplementary Figure S23). Additionally, the model trained with the Fourier-based prior has a better correlation of base importance to summit distance compared to L2-regularization ($-0.442 \pm 0.002$ vs. $-0.421 \pm 0.002$, $p < 1 \times 10^{-6}$ by Wilcoxon test), and a higher auPRC of ChIP-seq peak overlap with importance-ranked bases ($0.616$ vs. $0.585$) (Supplementary Figure S24). Importantly, the summit-distance correlation and peak-overlap auPRC were both *worse* than in models trained without any prior or regularization at all (without any prior/regularization, the summit-distance correlation is $-0.433 \pm 0.002$, and the auPRC of peak overlap is $0.595$). Even though L2-regularization may help improve the predictive performance more than the Fourier-based prior, it is detrimental to interpretability.

## 8.2 Fourier-based prior versus other attribution prior formulations

In a previous work, Erion et al. [9] proposed three formulations of attribution priors. We compare our Fourier-based prior to two of these three formulations: the smoothness prior and the sparsity prior (the third prior is defined on graphs, and is not applicable to sequence data) [9]. The smoothness prior penalizes the absolute difference in attributions between adjacent base pairs; the sparsity prior penalizes models for attribution uniformity, and rewards sparsity (i.e. high importance in few bases, and zero importance everywhere else) (Supplementary Methods Sec. 2.7).

We train binary SPI1 models with the smoothness prior or the sparsity prior, selecting a prior weight identically to how we picked the Fourier-based prior's weight (i.e. so that the prior loss is on a similar scale as the correctness loss). Compared to the Fourier-based prior, the smoothness and sparsity priors offer marginal improvements (if any) over the interpretability. The auPRC of peak overlap with importance-ranked bases is $0.616$ with the smoothness prior, $0.619$ with the sparsity prior, and $0.616$ with the Fourier-based prior. These minimal improvements in interpretability, however, come at the severe cost of poor predictive performance: over many random initializations, the validation-set prediction loss is significantly worse than the Fourier-based prior ($p < 5 \times 10^{-4}$ by a Mann–Whitney U-test, for both the smoothness and sparsity priors—see Supplementary Figure S25A). In fact, the smoothness prior hurts performance significantly more than when training with no prior at all.

Next, we tune the prior loss weight $\lambda$ for the smoothness and sparsity priors, and select the models with the lowest validation loss. After tuning, we somewhat rescue the validation-set prediction loss, although the Fourier-based prior still attains superior performance, *even without tuning its prior loss weight* (validation-set prediction loss is $0.268$ with the smoothness prior, $0.269$ with the sparsity prior, and $0.262$ with the Fourier-based prior). As a result, however, the smoothness and sparsity priors suffer in interpretability, and any edge over the Fourier-based prior vanishes: the peak-overlap auPRC of the smoothness and sparsity priors drop to $0.601$ and $0.597$, respectively, compared to $0.616$ for the Fourier-based prior (Supplementary Figure S25B).

Our observations are consistent with the intuition that the smoothness and sparsity priors *compete* with the correctness of the model. The smoothness prior penalizes *all* transitions in attributions (including transitions between motifs and background regions), thereby encouraging the model to find fewer motifs and to give them less importance. Similarly, the sparsity prior rewards models for finding as few motifs as possible (indeed, the sparsity prior loss is minimized when all bases have zero importance except one). The Fourier-based prior, however, does not penalize motifs at all, provided the transitions are somewhat smooth (smooth transitions are expected in many domains, including genomics, which have motifs with core and flanking sequence preferences). The Fourier-based prior can achieve its minimum value of 0 even when the model identifies many predictive motifs, with each being given high importance.

Taken together, these experiments indicate that compared to existing attribution prior formulations and traditional regularization (e.g. L2-regularization), only the Fourier-based prior is able to simultaneously maximize both interpretability *and* predictive performance.

## 9 Fourier-based priors resist reliance on GC content as a feature

The background GC content (i.e. proportion of G/C bases) of DNA sequences can be an informative feature to differentiate regulatory DNA from other genomic contexts, but reliance on it reduces the interpretability of genomic models, as it obfuscates the underlying motifs driving biological processes. Using simulated DNA sequences with varying amounts of GC bias in the positive training set, we show a simple binary SPI1 binding model trained with the Fourier-based prior resists relying on GC content as a feature, even when surrounding GC content becomes more and more informative (Figure 5). Notably, the prior is able to ignore the background GCs *without* attenuating the importance of specific GC-rich motifs (Supplementary Figure S26).

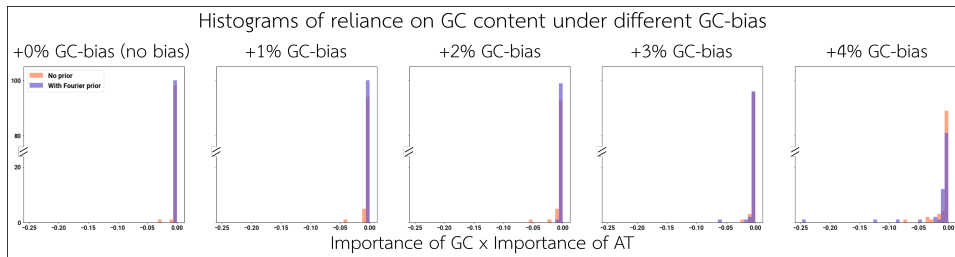

Figure 5: Over different levels of GC bias in the positive training set, we show how heavily models rely on distinguishing G/C and A/T in the background. Background GC content in the positive training set ranges from +0% (the same as the negative set) to an additive +4% more than the negative set. On a random sample of input sequences, the average product of G/C importance and A/T importance in the background sequence reflects how much the model relies on GC content as an informative feature (a more negative product implies a heavier reliance).

## 10 Conclusion

In this work, we introduced and demonstrated the application of a novel attribution prior based on Fourier transforms for improving the interpretability and stability of deep learning models on regulatory DNA. To our knowledge, this is the first application of attribution priors to deep learning models trained on genomic sequence, and the first application of a frequency-based attribution prior in any domain. Our method provides a direct way to train a model with interpretability for scientific discovery as an explicit goal. The Fourier-based prior remains flexibly applied to any architecture, but provides similar gains across models and prediction tasks. Notably, while our prior relied on input gradients to measure attributions during training time, all the benefits of the prior are clearly demonstrated using an alternative reference-based interpretation method (i.e. DeepSHAP). We note that anecdotally, the improvements in interpretability of the input gradients are even more dramatic, yet we relied on DeepSHAP scores at test-time due to their use of a biologically relevant background.

We showed that models trained with the Fourier-based prior have attributions with a significantly better signal-to-noise ratio, focusing importance primarily on biologically relevant motifs supported by independent data instead of irrelevant background sequence. Hence, models trained with the prior exhibit improved motif discovery and yield motif instance calls that are more likely to underpin regulatory function. Additionally, the Fourier-based prior improves the stability of model learning, directing models to consistently rely on regions of regulatory importance rather than irreproducibly learning background noise. Even when the models are trained with less data, they may be interrogated to discover motifs that are more similar to models that are trained with the full dataset.

Future work could explore other promising attribution prior formulations (e.g. penalizing Shannon entropy in the attributions) and harmoniously merge attribution priors like the Fourier-based prior with more standard forms of regularization. Advances in attribution priors will continue to improve the interpretability of deep learning models in genomics.

## Broader Impact

In this work, we focus on one primary application of deep learning models in genomics: to decipher the regulatory code of DNA by interpreting models trained on genome-wide molecular profiling experiments. These interpretations provide directed hypotheses for follow-up validation experiments [4], thereby improving our understanding of fundamental biology. Another important clinical application of these same models is to predict and interpret the molecular impact of DNA sequence mutations and variants in genomes of healthy and diseased individuals [2, 3, 5]. This clinical application can have significant positive impact in genomic medicine. However, issues regarding the stability and biological accuracy of the interpretations derived from these models have not been explored in a systematic manner.

Our work is one of the first attempts to addresses these issues by directly imposing a restriction on models at training-time, explicitly encouraging model learning to be more interpretable. Our method can be used in any genomic deep learning model being trained by backpropagation, where the drivers of genomic signal are motifs. This allows us to more confidently identify the sequence patterns that drive genomic regulation, including: 1) the discovery of protein-binding sequences; 2) the discovery of specific genome regulatory sequences; and 3) motif discovery and motif instance calling. This marks an important first step forward in the improvement of genomic deep learning models for scientific discovery.

In the future, we hope to extend this work to address the challenges associated with using these types of deep learning models for clinical genomic variant/mutation interpretation.

To our knowledge, this work is also the first to utilize Fourier transforms as an attribution prior. Although our application was limited to genomics, the expectation that input attributions should occur smoothly and in consecutive regions is prevalent in many other fields, as well. Additionally, Fourier analysis is a technique commonly employed in signal and image processing (e.g. frequency filtering). Thus, we also suspect that a Fourier-based attribution prior like the one described here may engender significant improvements in performance, stability, and interpretability in these other domains.

## Acknowledgments and Disclosure of Funding

The authors would like to thank Anna Shcherbina for her thoughtful advice and substantial technical support.

This work was supported by National Institutes of Health grants 1DP2GM123485-01 and 5U01HG009431-03. The authors declare no competing interests.

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
