[Supplementary Material 1 · supplementary_methods.pdf]

# Fourier-transform-based attribution priors improve the interpretability and stability of deep learning models for genomics: Supplementary Methods

All code is available at `https://github.com/amtseng/fourier_attribution_priors`

## 1 Training data preparation

### 1.1 SPI1 and GATA2 TF ChIP-seq

For these two transcription factors (TFs), we obtain data accessible through the ENCODE portal (`https://encodeproject.org/`), which has been processed using the ENCODE ChIP-seq pipeline [1, 2]. We select every TF ChIP-seq experiment with SPI1 or GATA2 as a target, satisfying the following conditions:

1. Experiment is of "released" status
2. Experiment has available unfiltered read BAMs aligned to hg38
3. Experiment has available BEDs of called peaks and IDR-filtered peaks aligned to hg38
4. Experiment has a matched control ChIP experiment with unfiltered read BAMs aligned to hg38
5. Cell type utilized in assay is not genetically modified

Based on these filtering criteria as of 4 Oct 2019, the ENCODE experiment IDs for SPI1 are ENCSR000BGQ, ENCSR000BUW, ENCSR000BIJ, ENCSR000BGW. The ENCODE experiment IDs for GATA2 are ENCSR000EVW, ENCSR000EWG, ENCSR000EYB. For each experiment, we obtain the unfiltered alignments, called peaks, and IDR-thresholded peaks, as available on the ENCODE portal (`https://encodeproject.org/`). For each TF, each experiment constitutes an output in the multi-task prediction models.

### 1.2 Nanog/Oct4/Sox2 TF ChIP-seq

Nanog, Oct4, and Sox2 TF ChIP-seq experiments in mouse ESCs were performed by Avsec et al. [3]. We use the stranded BigWig tracks (5' read counts) and IDR-thresholded called peaks as prepared by the authors. This constitutes three different experiments (i.e. prediction tasks). These reads and peaks are aligned to mm10.

### 1.3 K562 DNase-seq

We utilize a single DNase-seq experiment available through the ENCODE portal (`https://encodeproject.org/`): ENCSR000EOT [1, 2]. For this experiment, we obtain the unfiltered alignment BAMs, the set of called peaks that pass the ENCODE IDR filter, and the set of called peaks that did not pass the IDR filter. These files are obtained using the ENCODE ATAC-seq pipeline [4]. These reads and peaks are aligned to hg38.

Unlike TF ChIP-seq experiments, which have a matched control experiment, we perform bias correction by utilizing a control track that encodes the innate preferences of DNase [5]. We use the reads generated using this procedure, available as SRR1565781. These reads are processed into BigWigs using the ENCODE ATAC-seq pipeline into a single unstranded BigWig [4].

### 1.4 Reference genomes

We obtain reference genome Fasta files and chromosome sizes from the UCSC genome browser downloads, for hg38 and mm10.

### 1.5 Binary dataset preparation

The IDR-thresholded peaks are considered to be the high-confidence peaks. For the SPI1 and GATA2 TF ChIP-seq datasets, the 150,000 highest-scoring called peaks that do not overlap with an IDR-thresholded peak are considered ambiguous peaks, obtained using BEDtools v2.25.0 [6]. For the K562 DNase-seq dataset, the set of ambiguous peaks is generated by the ENCODE ATAC-seq pipeline [5]. For the Nanog/Oct4/Sox2 TF ChIP-seq datasets, there are no available ambiguous peak sets.

For all binary models, labels are generated by splitting the entire genome into 200 bp consecutive windows. A window is given a positive label if at least 100 bp of the window overlaps with a high-confidence peak, an ambiguous label if at least 100 bp of it overlaps with an ambiguous peak, and a negative label if neither of these two cases hold. For the TF ChIP-seq binary models, which have multiple tasks, the label generation procedure is performed independently for each task (i.e. each window will have an associated label for each task). These windows are then padded with 400 bp of context sequence on either side to form the final 1000 bp input to the network. This binary label generation was performed using seqdataloader [7].

## 1.6 Profile dataset preparation

For the SPI1 and GATA2 TF ChIP-seq datasets and the K562 DNase-seq dataset, we first merge together the unfiltered reads of all biological replicates in each experiment. For each experiment, we filter the merged reads using samtools v1.2 with flag 780, and keep only reads with quality at least 30 [8]. This is nearly identical to the ENCODE TF ChIP-seq and ATAC-seq pipelines [1, 4], except this retains duplicate reads, which may be useful in the prediction of profile shapes.

Using bedGraphToBigWig v4 [9], we convert these reads into BigWigs of 5' counts. For TF ChIP-seq datasets, the BigWigs are split into positive and negative strands, but for the K562 DNase dataset, the BigWig is unstranded. This gives each experiment a pair of—or a single—BigWig track(s) of 5' counts.

The profile 5'-count BigWigs for the Nanog/Oct4/Sox2 dataset were obtained from Avsec et al. [3] and are used as-is.

# 2 Model training

For both model architectures, we use a learning rate of 0.001. We train for a maximum of 20 epochs with early stopping (requiring an improvement in the validation loss by at least 0.001 over the last 3 epoch deltas). For binary models, we use a batch size of 128. For profile models, due to the larger model size, we use a batch size of 64. Batch size, learning rate, and early stopping criteria were selected by choosing values in previous works on similar architectures, and verifying that validation performance was comparable.

We also utilize reverse-complement augmentation in each batch, thereby effectively doubling the batch size.

Binary models learn a set of binary labels from a 1 kb input sequence. Profile models learn a set of 1 kb read profiles from a 1346 bp input sequence, as well as a set of scalar total read counts.

Models were trained using PyTorch 1.3.0, on Google Cloud using n1-standard-8 instances, each with an NVIDIA Tesla P100 GPU.

## 2.1 Training binary models

The binary model architecture consists of three consecutive 1D convolutions on the one-hot encoded DNA sequence. The convolutional layers have filter sizes of 15, 15, and 13, respectively (stride of 1). We do not use padding on the convolutional layers. For TF ChIP-seq datasets, we use 64 filters in each layer; for K562 DNase-seq models, we use 256. The convolutional layers have ReLU activations and batch normalization. The result of the convolutions are fed into a max-pooling layer of size 40 and stride 40. The max-pooling output is fed into two consecutive fully connected layers of size 50 and 15, respectively, both with ReLU activations. Finally, the result is passed to a final dense layer that outputs a sigmoid prediction. Each dense layer is also batch normalized.

The loss function is binary cross-entropy, averaged across the different tasks. The cross-entropy loss for the positive and negative classes are averaged in each batch. Each batch consists of genome bins where at least one task has a positive label, and an equal number of genome bins where no task had a positive label.

In each epoch, the network sees all genome bins where at least one task has a positive label. The "negative" samples are randomly subsampled at the beginning of each epoch.

## 2.2 Training profile models

The profile model architecture, adapted from Avsec et al. [3], has a profile output and a total counts output for each task. The architecture consists of seven consecutive 1D dilated convolutions on the one-hot encoded DNA sequence. The first dilated convolutional layer has a filter size of 21, and the subsequent layers have filter sizes of 3 (all have stride of 1). For TF ChIP-seq datasets, we use 64 filters; for K562 DNase-seq models, we use 256. Dilation size is 1 for the first layer (i.e. no dilation), and increases by powers of two in subsequent layers. The dilated convolutional layers have

ReLU activations. These dilated convolutional layers have summed residual connections, where the input to each layer is the sum of the outputs of all previous layers. Due to these summed connections, we utilize zero padding in these layers to ensure that the output of each layer is the same size. At the end, the final output of these dilated convolutional layers is cropped, cutting off any regions whose receptive field included a padded zero.

To compute the profile prediction, the last dilated convolution output is fed to another convolutional layer with kernel size 75 (stride 1) with no padding and no activation function (the output of this layer is length 1000). Finally, this result is then stacked with the control profile tracks, and fed to a final convolutional filter of size 1, such that the filter operates on one base of the logits and control profiles at a time. This constitutes the predicted profile logits.

To compute the total read count prediction, the last dilated convolution output is fed to a global average pooling layer, then to a dense layer. The result is concatenated with the total read counts in the control experiment, and fed through a final dense layer that predicts the log of the total read counts.

To compute the profile output and profile loss, the predicted profile logits are converted into probabilities by passing them through a softmax along the profile prediction dimension. This gives the predicted profile shape. For each track, these probabilities, along with the true read counts over the region, define a multinomial distribution, where each base is a bucket. The profile loss is the (negative) log probability of observing the true reads counts over this distribution, where the post-softmax predicted probabilities define the likelihood of a true read falling in each base. The results over all strands and tasks are averaged.

To compute the total counts loss, the predicted counts are treated as log counts, and the counts loss is simply the mean squared error of the log total counts, averaged over all strands and tasks (with a pseudocount of 1 for numerical stability).

When training, the profile loss and counts loss are given a weight of 1 and 20, respectively.

A positive example for a profile model is an input sequence and target profile track set centered at an IDR-thresholded peak summit (for any of the tasks). In each epoch, the network sees all IDR-thresholded peaks (the union over all tasks). Each batch also consists of an equal number of non-peak sequences/profiles, sampled uniformly at random from the genome every time a batch is generated. This implies that a sampled sequence intended for the negative set may overlap a peak, but note that not only is this unlikely, it also does not affect the correctness of any labels passed to the model. The peak sequences are also randomly jittered up to 128 bp from the summit in either direction, to augment the set of positive examples; this is done independently in each batch. For the K562 profile models, we do not train with a negative set due to time efficiency, as we anecdotally found that the profiles with random jitters were sufficient to yield good performance.

Our profile models also utilize control profiles for bias correction. For the SPI1, GATA2, and K562 models, these are matched controls. For the Nanog/Oct4/Sox2 profile models, the control is identical and multiplexed across all tasks.

Our TF ChIP-seq profile models (i.e. for SPI1, GATA2, and Nanog/Oct4/Sox2) are stranded, and predicted and control profiles have positive and negative strands. Our K562 DNase-seq profile models, however, are unstranded, and the predicted and control profiles are summed across both strands.

## 2.3 Fourier-based prior loss

In each batch during training, we only compute the Fourier-based prior loss for positive examples (i.e. binary examples where at least one task had a positive label, or profile examples originating from a peak). The Fourier-based prior loss is computed independently for each positive input sequence in the batch, and the loss is averaged.

The attributions $g(f, x)$ are computed for each positive example, resulting in a vector of the same length $N$ as the input sequence. In our models, $g(f, x)$ is based on the element-wise product of the input one-hot encoded sequence and the gradient of the output logits with respect to the input (as described in Shrikumar et al. [10]). This element-wise product is then summed over the base dimension, yielding a single score for each position in a given sequence. For binary models, we simply multiply the input sequence and the gradient of the binary logits with respect to the input sequence. For profile models, analogous to Avsec et al. [3], we use the profile prediction logits (pre-softmax), weighted by the post-softmax probabilities, and summed across the entire profile; these logit gradients are then multiplied by the input sequence to yield $g(f, x)$. For models with multiple tasks or strands, these logits are summed across these dimensions; thus, every positive training example will have a vector of attributions $g(f, x)$ that is the same length as the input sequence.

Note that $g$, our method of computing attribution priors, needs to be differentiable in order to include it in the attribution prior loss. Fortunately, for typical deep learning models (which are differentiable everywhere by construction), $g$ is also differentiable everywhere for most forms of backpropagation-based attribution methods (e.g. input gradients).

To compute the attribution prior loss, We take the absolute value of these attributions, and smooth them with a Gaussian kernel of standard deviation equal to 3 bp, cut to 1 standard deviation on each side (that is, the kernel has a width of 7 bp). The absolute-value and smoothing operations are intended to aid the prior in gracefully considering motifs with degenerate bases or variants (e.g. negative-scoring bases). Let the resulting smoothed attribution vector be $g^s(f, x)$. We then compute the discrete Fourier transform and recover the magnitudes $m^{(DC)}$ of the positive Fourier frequencies:

$$m^{(DC)} = FFT(g^s(f, x))$$

Note that $m^{(DC)}$ is a vector of length $\frac{N}{2} + 1$. We discard the component $m_0^{(DC)}$ which corresponds to "DC" (i.e. the average value of the attributions) to obtain $m$. $m$ is a vector of length $L = \frac{N}{2}$. We $\ell_1$-normalize the magnitudes $m$:

$$\hat{m} = \frac{m}{\sum\limits_{i=1}^{L} m_i}$$

Finally, we compute the attribution prior loss as the sum of the high-frequency normalized magnitudes:

$$w_i = \begin{cases} 1 & i \leq T \\ \frac{1}{1+(i-T)^s} & i > T \end{cases}$$

$$L_p(g(f, x)) = 1 - \sum_{i=1}^{L} w_i \hat{m}_i$$

We utilize a soft cut-off, with $s = 0.2$. This cut-off was selected by visualizing the graph of $h(x) = \frac{1}{1+x^s}$ and selecting a reasonable $s$ such that $h(x)$ decays gracefully over a span of roughly 50 bp (Supplementary Figure S1).

For a signal of length $N$, a rectangular pulse of size $p$ has a discrete Fourier transform of a sinc function with zeros every integer multiple of $\frac{N}{p}$. For binary models, the cut-off $T$ is selected to be 150 (in terms of frequency index), and for profile models, the cut-off is 200. In both cases, this corresponds to a motif length (i.e. pulse) of 6–7 bp (the input sequences to the binary and profile models have different lengths, and hence a different frequency index corresponds to a pulse of the same size). Note that this frequency threshold can be increased accordingly for datasets/tasks in which motifs are expected to be shorter than 6–7 bp, or vice versa.

When training with the Fourier-based prior, the prior loss is weighted by 1 for binary models. For profile models, we train a model first without the Fourier-based prior, then we take half of the validation loss (rounded to the nearest multiple of 5) after the model converges to be the weight of the prior loss (Supplementary Figure S4).

**Aside on the periodicity of the discrete Fourier transform**: Because the discrete Fourier transform implicitly constructs an infinitely long periodic signal from its input (i.e. by concatenating it end-to-end *ad infinitum*), there can sometimes be artifacts in the frequency domain. Fortunately, however, these issues are not typically present in our application, as the ends of the input sequence are generally devoid of any motifs or other informative features (and ergo the importance scores at the ends of the sequences are zero).

## 2.4 Peak subsampling

In some of our downstream analyses, we train models with only a subset of the dataset (e.g. 1% of the training set). To subsample our dataset, we limit the set of IDR-thresholded peaks to only the top 1% by signal strength. We do this by ranking all IDR-thresholded peaks across all tasks in descending order by signal strength, removing duplicate peaks (i.e. perfect overlaps), and retaining only the top 1% by count. This keeps only the 1% strongest/most confident peaks. For binary models, we recreate the training set from this new set of limited peaks. Negative bins are selected during training as usual. For profile models, we simply train with this smaller set of peaks as the positive set. Negative examples training are selected genome-wide, as usual.

For the Nanog/Oct4/Sox2 profile models trained on just 1% of the training data, we train for 80 epochs instead of the usual 20 (both with or without the Fourier-based prior).

## 2.5 Training logistics

In all models, we reserve chr1 as the test set, reserve chr8 and chr10 as the validation set, and partition all other chromosomes into the training set (for hg38 or mm10). Evaluation of a model on the validation and test set proceeds identically to the training set in terms of the selection of positive and negative examples (i.e. we utilize all positive examples in the validation/test set, and an equally sized random sample of negative examples).

To evaluate binary model performance, we examine the validation loss, and the test-set accuracy, auROC, and auPRC. These metrics are computed using balanced positive and negative classes. Because the auPRC is sensitive to class imbalance, we also compute an "estimated test auPRC", which estimates the true auPRC on the full test set as if the negative examples were not subsampled to achieve a balanced set. We use this estimate rather than computing the auPRC on the full test set for computational efficiency. This estimate is calculated by artificially inflating the false positive rate, with the assumption that the negatives already present are representative of the full distribution of negatives.

To evaluate profile model performance, we examine the profile loss (i.e. the negative log-likelihood of the profiles).

For each dataset and model architecture, we train models over 30 different random initializations. In our downstream analyses (unless otherwise stated), we always consider the model with the best validation loss over all random initializations and epochs. For profile models, we compare the profile validation loss, rather than the aggregate loss of the profiles and counts outputs.

## 2.6  L2-regularization models

We train SPI1 binary models with L2-regularization (without the Fourier-based prior), using an added L2-norm loss, consisting of the L2-norm of all trainable parameters in the network.

We tune the L2-norm loss weight over 25 logarithmically sampled random weights from $10^{-8}$ to $10^2$, eventually settling on the optimal weight of 0.0001, which yielded the lowest validation loss. Using this optimal L2 loss weight, we train 20 random initializations of SPI1 binary models. In all analyses, we use the model/epoch with the best validation loss.

## 2.7  Smoothness prior and sparsity prior

We train SPI1 binary models with the smoothness prior or the sparsity prior (without L2-regularization or the Fourier-based prior), as described in Erion et al. [11]. Let $g(f, x)$ be the vector of attributions, and $N$ the vector's length, as described in Sec. 2.3. We take the absolute value of the attributions to obtain the non-negative vector $g^a(f, x)$. Let $g^a(f, x)_i$ be the $i$th element of $g^a(f, x)$.

Although the smoothness prior is defined in Erion et al. [11] to penalize differences in attributions between adjacent pixels in images, we adapt it for sequence inputs as follows:

$$L_p(g(f,x)) = \sum_{i=1}^{N-1} |g^a(f,x)_i - g^a(f,x)_{i+1}|$$

The sparsity prior is defined as:

$$L_p = -\frac{\sum_{i=1}^{N}\sum_{j=1}^{N} |g^a(f,x)_i - g^a(f,x)_j|}{N\sum_{i=1}^{N} g^a(f,x)_i}$$

For each prior, we train models over 15 random initializations. We select the prior loss weight $\lambda$ so that the ratio of $L_c$ to $\lambda L_p$ is roughly the same as the ratio for a trained SPI1 binary model using the Fourier-based prior ($L_c$ is the model's correctness loss, which in this case is binary cross entropy). This ensures that the relative magnitudes of $L_c$ and $\lambda L_p$ are roughly the same. Using this method, we select a prior loss weight of 20000 for the smoothness prior, and 0.01 for the sparsity prior. We pick the models with the best validation loss for downstream analyses where we consider models without a tuned prior loss weight.

We then further tune the prior loss weight for the smoothness and sparsity priors. For the smoothness prior, we tune over 15 logarithmically sampled random weights from $10^{-3}$ to $10^7$. For the sparsity prior, we tune over 15 logarithmically sampled random weights from $10^{-4}$ to $10^2$. The best prior weights were roughly 5000 for the smoothness prior, and 0.003 for the sparsity prior. We pick the models with the best validation loss for downstream analyses where we examine models after tuning the prior weight.

### 2.8 Models on simulated sequences

We train single-task SPI1 binary models with simulated DNA sequences for the positive and negative labels. We also only train for 5 epochs, where each epoch has the same number of positive and negative examples as a single-task SPI1 binary model trained on ENCSR000BGQ. All other training details are identical to the SPI1 binary models on real experimental data.

To create the input sequence for a negative label, we synthesize a sequence where every base is sampled independently and uniformly from A, C, G, and T. To create the input sequence for a positive label, we similarly sample a sequence from a uniform distribution of bases, and subsequently place a single instance of the SPI1 motif in the center. The placed motif is sampled from a Position Frequency Matrix (PFM) that was generated using HOMER 2 [12] on the set of IDR-thresholded peaks for SPI1, aggregated over all 4 ENCODE experiments (Supplementary Figure S21). Specifically, we used findMotifsGenome.pl in hg38, with a length of 12 and size of 200. We keep only the top motif. We then trim the motif by removing flanking regions with less than 20% of the information content of the base with the highest information content in the motif. Positive and negative sequences are sampled randomly at training-time every time a batch is generated. We use simdna to generate simulated positive and negative sequences [13].

We train models with and without the Fourier-based prior over 3 random initializations each, and pick the model and epoch with the lowest validation loss for all downstream analyses.

### 2.9 Comparison to Basset

We compare the predictive performance of our single-task K562 binary models to Basset [14]. We download the model architecture and weights from Kipoi, a repository of predictive models for genomics [15]. We then use this restored Basset model to evaluate our K562 binary test set (Sec. 1.5), using all positive bins and an equally sized random sample of the negative bins. We compute the auROC and auPRC of the predictions for Basset, as well as our best-performing single-task K562 binary model (trained without any prior).

Note that while our binary models accept input sequences of 1000 bp, Basset takes in input sequences of 600 bp. When we test on Basset, we create our test set in the same way as described in Sec. 1.5, except we pad the sequence to length 600 bp rather than 1000 bp (i.e. the central 200 bp is still required to overlap a high-confidence peak by at least 100 bp to be considered a positive).

Additionally, Basset is trained on hg19, whereas our dataset is created using hg38 annotations. Because the inputs to the models are simply one-hot encoded sequences, however, it is expected that there is virtually no difference in the input data distributions.

## 3 DeepSHAP computation

To compute attributions (i.e. importance scores) for input sequences, we utilize DeepSHAP [16]. For binary models, we explain the binary prediction logits, summed across tasks. For profile models, we explain the profile prediction logits weighted by the final post-softmax probabilities, summed across the profile, and summed across tasks and strands.

Our baseline/reference set for DeepSHAP computation consists of 10 randomly shuffled versions of the input sequence, preserving dinucleotide frequencies. This choice of reference was recommended for genomic sequences in the DeepLIFT paper [17].

In subsequent sections, we use the term "hypothetical" DeepSHAP importance to refer to the estimated importance scores of an input sequence for all possible bases (i.e. the estimated DeepSHAP attributions for each base if it were present, hypothetically, at that location). The procedure for computing "hypothetical" importance scores can be found in Shrikumar et al. [18]. We use the term "actual" DeepSHAP importance to refer to the hypothetical importance multiplied by the input sequence (i.e. projected onto the bases that are actually in the sequence) and reduced to a single dimension along the input sequence by summing over the base identities at each position.

In order to make DeepSHAP work with PyTorch models and to easily produce "hypothetical" importance scores, we used a slightly modified version of the DeepSHAP library, available at `https://github.com/amtseng/shap`.

## 4 Signal-to-noise ratio of attributions

For each dataset, we select 1000 random positive examples from the test set and examine their interpretability. For profile models, we include a random jitter of $\pm 128$ bp to avoid center-bias. We compute the "actual" DeepSHAP

importance scores as described above for each sequence and quantify each sequence's signal-to-noise ratio as the sum of the high-frequency normalized Fourier components, as well as the Shannon entropy. For simplicity, the sum of high-frequency normalized Fourier components is computed identically to the value of the Fourier-based prior loss (Sec. 2.3), but with $s = \infty$ (i.e. no softness in the frequency cut-off). The Shannon entropy is computed by taking the absolute value of the attributions and normalizing them along the sequence to pseudo-probabilities. We report the average high-frequency Fourier sum and the average Shannon entropy over the sampled test-set sequences, as well as the standard error of the mean.

We use this same procedure to compare the attributions of our SPI1 binary models trained with L2-regularization versus the Fourier-based prior.

## 5 Motif discovery and motif calling

For our Nanog/Oct4/Sox2 models, we call motifs using TF-MoDISco v0.5.5.5 [18]. On the entire set of IDR-thresholded peaks in the test set, we compute DeepSHAP importance scores on the individual tasks for each TF, and run TF-MoDISco. We utilize a sliding window size of 21, flank size of 10, and seqlet FDR of 0.01. From the resulting motif clusters, we select only the motifs that have an average information content of at least 0.6 over some window of length 6. We then trim the motifs by removing flanking regions with less than 20% of the information content of the base with the highest information content in the motif. We also remove motifs corresponding to homopolymer repeats. For binary models, due to their innate limitations in distinguishing primary motifs, we further filter motifs to have at least 750 supporting seqlets. For each motif, this gives us a PWM (Position Weight Matrix) and a CWM (Contribution Weight Matrix). The CWM consists of the aggregated "actual" DeepSHAP importance scores [3]. The background frequencies used to compute the PWM are 27% A or T, and 23% G or C.

To call motif instances, we select 1000 random positive examples from the test set. For profile models, we include a random jitter of $\pm128$ bp to avoid center-bias. For each motif, we compare it to every possible window in these sequences, and call a motif instance if:

1. The PWM match score is positive
2. The summed continuous Jaccard similarity (as defined below in Sec. 7) between the motif CWM and the underlying "actual" DeepSHAP scores of the sequence window is in the top 10% for that motif

This method of calling motif instances was adapted from Avsec et al. [3], which found that using CWMs to call motifs was significantly more effective for identifying high-confidence motifs (as supported by the experimental assay) compared to traditional PWM scanning.

We then rank the motif calls by the total DeepSHAP importance magnitude, summed across the instance, and compute precision–recall for which motif instances overlap with corresponding ChIP-nexus peaks by at least 1 base.

## 6 Performance on binary models

On our binary models, we compute accuracy, auROC, and auPRC on the test set, randomly subsampling negative bins to achieve balanced positives and negatives. We also compute an estimate of the auPRC on the imbalanced test set by artificially inflating the false positive rate to simulate auPRC computation on the entire test set. These are computed over all 30 random initializations for each dataset. The performance metrics of the best-performing model over the random initializations is recorded, with the best-performing model being the model with the lowest validation loss over all random initializations and epochs.

## 7 Stability of model learning

For each dataset, we select 100 random positive examples from the test set and examine the consistency of their "hypothetical" DeepSHAP importance across different models. For profile models, we include a random jitter of $\pm128$ bp to avoid center-bias.

For each input sequence, we compute the similarity of "hypothetical" DeepSHAP importance from any two models using the continuous Jaccard similarity metric [18]. The continuous Jaccard similarity is computed between the two sets of $\ell_1$-normalized "hypothetical" attribution scores for each base, and the resulting similarities are summed across the length of the sequence. For two $d$-vectors $u$ and $v$ (here, $d = 4$ for the 4 bases), the Jaccard similarity per base is

computed as follows:

$$\sum_{i=1}^{d} \text{sign}(u_i)\text{sign}(v_i)\frac{\min\{|u_i|,|v_i|\}}{\max\{|u_i|,|v_i|\}}$$

To quantify the similarity across all 30 random initializations of a model, for each of the sampled input sequences, we compute the pairwise continuous Jaccard similarity sum for each pair of models and average the scores across all $\binom{30}{2}$ pairs. To quantify the similarity between models trained with all versus only 1% of the training data, we select the top 5 best-performing models when trained with the entire training set, and the top 5 best-performing models when trained with only 1% of the training set, and we compute the pairwise continuous Jaccard similarity sum across all $5 \times 5$ pairs between the two conditions. The best-performing models are selected by picking the 5 models (i.e. 5 different random initializations) in each condition with the best validation loss (or profile validation loss, for profile models). We report the average similarity over the sampled test-set sequences (where the similarity value for each sequence is the average continuous Jaccard score over many pairs of models), as well as the standard error of the mean.

## 8 Reliance on biologically relevant regions

For each dataset, we select 1000 random positive examples from the test set and examine the relationship between the "actual" DeepSHAP attributions and the underlying summits/peaks. For profile models, we include a random jitter of $\pm 128$ bp to avoid center-bias.

First, over each of the 1000 sampled sequences, we compute the Spearman correlation of the magnitude of the "actual" DeepSHAP importance at each base to the distance to the closest IDR-thresholded peak summit. We report the average Spearman correlation over the sampled test-set sequences as well as the standard error of the mean.

Next, over all 1000 sampled sequences in aggregate, we rank all bases in descending order by the magnitude of "actual" DeepSHAP importance, and ask whether the top bases overlie IDR-thresholded peaks. We use this to compute precision–recall curves, where thresholds are the "actual" DeepSHAP magnitude, and a positive is when a base overlies an IDR-thresholded peak.

We use this same procedure as above to compare the attributions of our SPI1 binary models trained with L2-regularization versus the Fourier-based prior, and when comparing the Fourier-based prior with the smoothness and sparsity priors.

For the K562 models and Nanog/Oct4/Sox2 models, we perform further analyses using orthogonal footprints or ChIP-nexus data, respectively. The K562 footprints are computed by Vierstra et al. [19]. From their published data, we aggregated the footprints of all K562 experiments using BEDtools merge [6]. For the Nanog/Oct4/Sox2 models, in addition to ChIP-seq experiments, Avsec et al. [3] also performed ChIP-nexus experiments, and we utilize their IDR-thresholded peaks from the ChIP-nexus experiments.

To compute the fraction of importance in a ChIP-nexus peak or footprint, we take the absolute value of the "actual" DeepSHAP importance of each of the sampled sequences, and calculate the proportion (out of the total sum across that sequence) within a ChIP-nexus peak or DNase footprint. We also repeat the rank-based analysis with footprints or ChIP-nexus peaks instead of ChIP-seq or DNase-seq peaks. We report the average proportion over the sampled test-set sequences as well as the standard error of the mean.

For some of the binary models, we also compute the motif clusters of the high-importance regions that the Fourier-based prior highlighted, but where the regions did not overlie a ChIP-nexus peak or a footprint. Over the same sample of 1000 test sequences used to compute the auPRC of ChIP-nexus peak or footprint overlap, we take the top 200,000 most important bases (i.e. highest magnitude of DeepSHAP score) for which the base did not overlie a ChIP-nexus peak or a footprint. For each base, we expand to a centered 50 bp region, discarding overlaps (keeping the higher-ranked bases). For the resulting regions, we run the TF-MoDISco v0.5.5.5 [18] clustering algorithm only, clustering seqlets of length 30 into motifs.

To evaluate the ability of our models trained on simulated data to place importance only on motifs, we employ a similar procedure as above. We take a random sample of 100 simulated positive sequences (i.e. with placed motifs). For each sequence, we identify instances of the SPI1 motif by considering locations where the match score to the SPI1 PWM (i.e. the PWM used to construct the sequences) is over 0.9. The match score of a potential motif instance is computed as the sum of the entries in the PWM corresponding to the bases in the instance (i.e. total log-odds). Here, we use a uniform background (i.e. 25% A, C, T, or G) to compute the PWM. To compute the fraction of importance in the motifs, we use the same procedure as above to compute fraction of importance in a ChIP-nexus peak or footprint, but use motif instances instead of peaks/footprints. To compute the auPRC of base overlap with motif instances, we use the same

procedure as above to compute precision–recall of base overlap with peaks/footprints, but use called motif instances instead.

## 9 GC content simulations

We train single-task binary SPI1 models on varying amounts of GC bias. The procedure for constructing simulated sequences is identical to the procedure described above for training single-task binary SPI1 models without GC bias. The only difference is that when we synthesize a positive sequence (i.e. with an inserted motif instance), the background sequence can have a higher amount of GC content. The negative sequence set always has sequences of equal GC and AT content (i.e. no bias). We train models on 5 different levels of GC content: +0%, +1%, +2%, +3%, and +4%. A level of GC bias of $+x\%$ means the probability of G or C in the positive background sequences is $(50 + x)\%$, while the negative sequence background has a G/C probability of 50% (i.e. no bias at all).

We train models with and without the Fourier-based prior over 3 random initializations each, and pick the model and epoch with the lowest validation loss for all downstream analyses.

To quantify how much a model is relying on GC content in the background, we sample 100 random positive sequences with the specified amount of GC bias, and compare the importance of A or T to the importance of G or C. For each sequence, we first mask out all instances of the SPI1 motif by ignoring any locations where the match score to the SPI1 PWM (i.e. total log-odds) is over 0.9. When scanning each simulated sequence for matches to the SPI1 motif, we compute the PWM using background frequencies that match the GC content of the sequence. We then consider the (signed) "hypothetical" DeepSHAP importance of A or T versus G or C, normalized by the maximum importance over the entire "hypothetical" importance track. We compute the product of A/T importance and G/C importance per base, and average over each input sequence.

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

[Supplementary Material 2]

# Fourier-transform-based attribution priors improve the interpretability and stability of deep learning models for genomics: Supplementary Figures and Tables

Figure S1: Weight applied to Fourier components in attribution prior loss. To compute the Fourier-based attribution prior loss, the Fourier components corresponding to positive frequencies of the attributions are weighted (these weights $w$ are plotted here). This weighted sum constitutes the score of the attributions, and 1 minus this score becomes the attribution prior loss value. Note that because input sequences to binary and profile models have different lengths, the lengths of the discrete Fourier transform components are also different; in both cases, the frequency threshold $T$ corresponds to a minimum expected motif length of 6–7 bp.

Figure S2: Schematic of binary model architecture. A one-hot encoded sequence is fed into three consecutive convolutional layers. The resulting activations are passed through a max pooling layer, followed by three dense layers, where the final dense layer outputs a sigmoid-transformed binary prediction.

Figure S3: Schematic of profile model architecture, based on the architecture in Avsec et al. [1]. A one-hot encoded sequence is fed into six consecutive dilated convolutional layers with summed residual connections. For each task, the model predicts a profile shape and a read count. The profile shape prediction is obtained by feeding the activations from the dilated convolutions to another convolutional layer with a large kernel size, concatenating the result with a set of control profiles, and performing a length-1 convolution over the concatenation to yield a profile shape prediction. The read count prediction is obtained by feeding the activations from the dilated convolutions through a global average pooling layer, concatenating the result with a set of control read counts, and passing this concatenation through a dense layer to obtain predicted read counts.

Table S1: Predictive performance of single-task K562 binary model versus Basset

|        | Basset | Single-task model |
|--------|--------|-------------------|
| auROC  | 0.838  | **0.966**         |
| auPRC  | 0.839  | **0.964**         |

We compare our single-task K562 binary models (trained without any prior—see Supplementary Figure S2) to Basset, a massively multi-tasked binary model trained to predict chromatin accessibility in 164 cell types (including K562) [2]. On our K562 test set, our single-task model achieves better predictive performance than the K562 output head of Basset.

Figure S4: Training and validation correctness loss trajectories. For each architecture and dataset, we show the trajectory of the training and validation correctness losses (i.e. excluding any attribution prior loss) after each epoch of training, over all random initializations. Loss values are plotted beginning after the first epoch of training. In general, binary models (left) overfit very easily, with validation loss visibly growing after the first epoch. Profile models (right), however, are much more resilient to overfitting, as they benefit from extensive data augmentation through random jitters in the input sequences. On binary models (left), red circles indicate the models that achieve the lowest validation loss, trained with and without the Fourier-based prior.

Table S2: Improved interpretability on test-set sequences

| | | Average sum of high-frequency Fourier components | | Average entropy | |
|---|---|---|---|---|---|
| | | No prior | With prior | No prior | With prior |
| Binary | SPI1 | $0.416 \pm 0.002$ | $\mathbf{0.374 \pm 0.002}$ | $8.35 \pm 0.01$ | $\mathbf{7.89 \pm 0.01}$ |
| | GATA2 | $0.442 \pm 0.002$ | $\mathbf{0.394 \pm 0.002}$ | $8.55 \pm 0.01$ | $\mathbf{7.96 \pm 0.02}$ |
| | Nanog/Oct4/Sox2 | $0.449 \pm 0.002$ | $\mathbf{0.381 \pm 0.002}$ | $7.98 \pm 0.01$ | $\mathbf{6.76 \pm 0.02}$ |
| | K562 | $0.494 \pm 0.002$ | $\mathbf{0.465 \pm 0.002}$ | $8.84 \pm 0.01$ | $\mathbf{8.65 \pm 0.01}$ |
| Profile | SPI1 | $0.449 \pm 0.002$ | $\mathbf{0.381 \pm 0.002}$ | $7.98 \pm 0.01$ | $\mathbf{6.76 \pm 0.02}$ |
| | GATA2 | $0.449 \pm 0.002$ | $\mathbf{0.416 \pm 0.001}$ | $8.49 \pm 0.01$ | $\mathbf{7.34 \pm 0.02}$ |
| | Nanog/Oct4/Sox2 | $0.564 \pm 0.001$ | $\mathbf{0.452 \pm 0.001}$ | $9.01 \pm 0.01$ | $\mathbf{8.01 \pm 0.02}$ |
| | K562 | $0.567 \pm 0.001$ | $\mathbf{0.452 \pm 0.001}$ | $9.20 \pm 0.01$ | $\mathbf{8.80 \pm 0.01}$ |

Figure S5: Signal-to-noise ratio of attributions across test-set sequences. For each architecture and dataset, we compute DeepSHAP importance scores for 1000 randomly selected peak sequences from the test set. An improvement in the signal-to-noise ratio of the attributions is quantified as a reduction in the high-frequency Fourier component magnitudes, and as a reduction in Shannon entropy. We compare the sum of normalized high-frequency Fourier components and the Shannon entropy for each sequence, between models trained with versus without the Fourier-based prior.

Figure S6: Specific examples of improved interpretability in DeepSHAP scores. For each architecture and dataset, we show the DeepSHAP attribution scores of specific peak sequences. For each selected input sequence, we display the value of the DeepSHAP importance for the bases present along the entire input sequence, as well as the base-pair-level attributions in the summit region.

Figure S7: Discovered motifs from Nanog/Oct4/Sox2 binary models. For each TF, we use TF-MoDISco to discover motifs using DeepSHAP attributions of test-set peak sequences. We show the relevant motifs identified by TF-MoDISco which pass our thresholds (Supplementary Methods Sec. 5), and match them to known motifs from Avsec et al. [1]. "(RC)" denotes that the motif shown is reverse-complemented relative to the orientation in Avsec et al. [1]. The number in the top left of each motif indicates the number of seqlets identified by TF-MoDISco that underlie the motif.

Figure S8: Discovered motifs from Nanog/Oct4/Sox2 profile models. For each TF, we use TF-MoDISco to discover motifs using DeepSHAP attributions of test-set peak sequences. We show the relevant motifs identified by TF-MoDISco which pass our thresholds (Supplementary Methods Sec. 5), and match them to known motifs from Avsec et al. [1]. "(RC)" denotes that the motif shown is reverse-complemented relative to the orientation in Avsec et al. [1]. The number in the top left of each motif indicates the number of seqlets identified by TF-MoDISco that underlie the motif. While we focus our downstream analysis on the best-performing models with and without the Fourier-based prior, the best-performing profile model trained without the prior identified much fewer motifs than the model trained with the prior, so we also show the motifs identified using TF-MoDISco on the second-best-performing models, both with and without the prior. This demonstrates that the profile models without the Fourier-based prior are capable of learning the larger set of motifs expected, but are limited by noisy and irreproducible attributions.

Figure S9: Motif instance call support. For each TF in the Nanog/Oct4/Sox2 models, we perform motif instance calling using the discovered motifs on a sample of 1000 test-set peak sequences. We rank the motif instance calls by total DeepSHAP importance, and compute a cumulative count of how many instances overlap with a ChIP-nexus peak for that TF. Note that the models trained without the prior typically have fewer motif calls in total (due to lower-quality attribution scores and fewer motifs discovered by TF-MoDISco), resulting in the shorter red lines.

Figure S10: Case study of the "Nanog (alt)" motif in Nanog/Oct4/Sox2 profile models. TF-MoDISco identified the GGAAAT "Nanog (alt)" motif from the profile Nanog/Oct4/Sox2 model trained with the Fourier-based prior, but not from the model trained without the prior, even on the Nanog prediction task specifically (Supplementary Figure S8). Focusing on the Nanog prediction task, we show examples of sequences where TF-MoDISco identified a Nanog (alt) motif from the model trained with the prior, and show the importance scores of the same sequence from the model without the prior. We show both the DeepSHAP importance scores, and the perturbation scores derived from *in silico* mutagenesis (ISM). The highlighted regions indicate the location of the Nanog (alt) motif. Notably, ISM scores from the model trained without the prior are generally noisier compared to the model trained with the prior. More importantly, this demonstrates that the model trained without the prior is learning the Nanog (alt) motif, but it is not visible from DeepSHAP importance scores. The model trained with the Fourier-based prior, however, clearly highlights this motif using both methods of interpretation (i.e. DeepSHAP and ISM). This indicates that the Fourier-based prior allows the model to reveal its learned motifs in a human-interpretable way, especially when it is too computationally expensive to rely on perturbation-based scoring methods like ISM, which take orders of magnitude longer to run than DeepSHAP.

Figure S11: Validation loss and test-set performance of binary models. For each dataset, we consider the validation- and test-set performance of models trained with and without the Fourier-based prior, over 30 random initializations each. Validation loss is computed over all positive examples in the validation set and a equal-sized sample of negative validation examples. Test accuracy, test auROC, and test auPRC are computed over all positive examples in the test set and a equal-sized sample of test negative examples. "Estimated test auPRC" is an estimated measure of auPRC on the full test set without subsampling the negative examples, and is computed by artificially inflating the false positive rate (Supplementary Methods Sec. 2.5).

Figure S12: Validation loss and test-set performance of binary models on sparse training sets. For each dataset, we consider the validation- and test-set performance of models trained with and without the Fourier-based prior (on only 1% of the training set), over 30 random initializations each. Validation loss is computed over all positive examples in the validation set and a equal-sized sample of negative validation examples. Test accuracy, test auROC, and test auPRC are computed over all positive examples in the test set and a equal-sized sample of test negative examples. "Estimated test auPRC" is an estimated measure of auPRC on the full test set without subsampling the negative examples, and is computed by artificially inflating the false positive rate (Supplementary Methods Sec. 2.5)

Figure S13: Stability of DeepSHAP scores across different models on test-set sequences. For each architecture and dataset, we sample 100 peak sequences from the test set and compute the DeepSHAP attributions for the sequence in multiple models. For each sequence, we compute the pairwise similarity of the attributions between 30 random initializations (left), or between the top 5 models trained with all of the data versus the top 5 trained with only 1% of the data (right). We quantify attribution similarity by computing the continuous Jaccard score at each base, and summing the scores across the sequence.

Figure S14: Specific examples of DeepSHAP attribution stability across different random initializations (binary models). For each binary model, we show an example of the DeepSHAP attributions on a test-set sequence between a pair of models of different random initializations, comparing models trained with versus without the Fourier-based prior.

Figure S15: Specific examples of DeepSHAP attribution stability across different random initializations (profile models). For each profile model, we show an example of the DeepSHAP attributions on a test-set sequence between a pair of models of different random initializations, comparing models trained with versus without the Fourier-based prior.

Figure S16: Specific examples of DeepSHAP attribution stability between training on all versus 1% of the training set. For each architecture and dataset, we show an example of the DeepSHAP attributions on a test-set sequence between a model trained on all and only 1% of the training set, comparing models trained with versus without the Fourier-based prior.

Figure S17: Precision–recall of importance-ranked base overlap with called peaks. For each architecture and dataset, we sample 1000 peak sequences from the test set and compute the DeepSHAP attributions. We rank bases in descending order of total importance, and generate a precision–recall curve by treating the set of bases that overlap an underlying ChIP-seq or DNase-seq peak as "positives". See Table 3 in the main text for the corresponding auPRC values.

Figure S18: Fraction of importance in ChIP-nexus peaks or DNase footprints. For Nanog/Oct4/Sox2 TF ChIP-seq or K562 DNase-seq models, we sample 1000 peak sequences from the test set and compute the fraction of total DeepSHAP importance that overlaps a Nanog/Oct4/Sox2 ChIP-nexus peak or K562 footprint. For each input sequence, we compare the proportion of attribution by magnitude overlapping a ChIP-nexus peak or footprint when a model is trained with versus without the Fourier-based prior. For Nanog/Oct4/Sox2 models, we show the importance overlap using the total importance across all three tasks (in which case we use the union of ChIP-nexus peaks to define overlapping regions), as well the overlap using the importance scores of each individual task (in which case we use only the ChIP-nexus peaks of the corresponding task). See Table 4 in the main text for the corresponding average overlap values.

Figure S19: Precision–recall of important bases overlapping with ChIP-nexus peaks or DNAse footprints. For models trained to predict Nanog/Oct4/Sox2 TF ChIP-seq or K562 DNase-seq, we sample 1000 peak sequences from the test set and compute the DeepSHAP importance. We rank bases in descending order of total importance and generate a precision–recall curve by treating the set of bases that overlap a Nanog/Oct4/Sox2 ChIP-nexus peak or K562 DNase-seq footprint as "positives". For Nanog/Oct4/Sox2 models, we show these curves both using a ranking based on the total importance across all three tasks (in which case we use the union of ChIP-nexus peaks over the three tasks for the labels), as well as a ranking based on the importance scores of each individual task (in which case we use only the ChIP-nexus peaks of the corresponding task). See Table 4 in the main text for the corresponding auPRC values.

Figure S20: Motifs in important regions outside of ChIP-nexus peaks or footprints. For the Nanog/Oct4/Sox2 TF ChIP-seq and K562 DNase-seq binary models, we examine regions where models trained with the Fourier-based prior place high DeepSHAP importance, yet do not overlap Nanog/Oct4/Sox2 ChIP-nexus peaks or K562 footprints, respectively. **A)** We cluster these regions using the TF-MoDISco clustering algorithm, and show the PWMs of top motif clusters, along with annotations of relevant TFs that are associated to each motif. **B)** We show some specific examples where a model trained with the Fourier-based prior places higher importance (relative to the model trained without the prior) outside of a ChIP-nexus peak or K562 footprint. Yellow shading denotes the location of the ChIP-nexus peak (left) or K562 footprint (right). This illustrates the Fourier-based prior's highlighting biologically relevant motifs outside of peak or footprint regions; the model trained without the prior, on the other hand, identifies these motifs more weakly/noisily, or not at all. Several mechanisms exist by which secondary motifs outside the central peak region can nonetheless assist TF binding within the peak region (e.g. via cofactor interactions [3] or 1D sliding [4]). A binary prediction model would be correct in identifying such motifs as being predictive of peak strength. Note that in contrast to binary models, profile models are less likely to detect such secondary motifs, as these motifs contribute to peak strength without contributing to the shape of the peak itself (peak shape is primarily dictated by motifs lying within the central peak region).

Figure S21: SPI1 motif used in constructing simulated sequences. We show the SPI1 motif (left) and its reverse complement (right) used to simulate SPI1-binding sequences for models trained on simulated data. Shown here are (from top to bottom): the Position Frequency Matrix (PFM) of the top motif identified by running HOMER 2 on IDR-thresholded peaks for SPI1 (Supplementary Methods Sec. 2.8); the trimmed motif PFM after removing flanks with low information content; and a PWM of the trimmed motif derived from weighting the PFM by information content.

Figure S22: Attributions in simulated motif instances. We examine binary models trained to predict single-task SPI1 binding on simulated sequences. On a random sample of 100 motif-containing sequences, we compare models trained with versus without the Fourier-based prior by computing: **A)** the proportion of DeepSHAP importance overlying a motif instance; and **B)** the precision–recall of importance-ranked bases overlying motif instances. **C)** We also select an example sequence, and show the DeepSHAP attributions; the model trained with the Fourier-based prior cleanly highlights all three motif instances. Of the two zoomed-in attribution score tracks in Panel **C**, the upper track represents the hypothetical importance scores (i.e. the importance that would be given to each of the four bases at each position, even if that base were not in the input sequence—see Supplementary Methods Sec. 3), and the lower track represents the actual importance scores. See Supplementary Figure S21 for the SPI1 PWM used in the simulations. Note that the central motif instance is in the reverse-complement orientation.

Figure S23: Interpretability of attributions with L2-regularization versus Fourier-based prior. For a SPI1 binary model, we show the DeepSHAP attributions for 1000 randomly selected peak sequences from the test set. We compare the sum of normalized high-frequency Fourier components and the Shannon entropy for each sequence, between a model trained with L2-regularization (i.e. weight decay) and a model trained with the Fourier-based prior. For a single selected test peak, we also display the value of the DeepSHAP importance for the bases present along the entire input sequence, as well as a zoomed-in view of a region close to the peaks summit. We also show the "hypothetical" attributions along each input sequence (i.e. the importance that would be given to each of the four bases at each position, even if that base were not in the input sequence—see Supplementary Methods Sec. 3).

Figure S24: Precision–recall of important bases overlapping with ChIP-seq peaks (L2-regularization versus Fourier-based prior). For a SPI1 binary model, we sample 1000 peak sequences from the test set and compute the DeepSHAP attributions. We rank bases in descending order of total importance and generate a precision–recall curve by treating the set of bases that overlap a SPI1 ChIP-seq peak as "positives". Shown here are the precision–recall curves for models trained with L2-regularization versus the Fourier-based prior.

Figure S25: Comparison of predictive performance and interpretability between the Fourier-based prior and alternative attribution prior formulations. **A)** On SPI1 binary models trained with the smoothness prior or sparsity prior [5], we show the distribution of the predictive performance (as measured by validation-set prediction loss) compared to models trained with the Fourier-based prior or with no prior at all, over several random initializations. The predictive performance of the smoothness and sparsity priors is significantly worse than the Fourier-based prior. Models trained with the smoothness prior have worse performance than models trained with no prior at all. **B)** We tune the prior loss weights for the smoothness and sparsity priors, and select the models with the lowest validation loss (Supplementary Methods Sec. 2.7). In these models, the smoothness and sparsity priors have poor interpretability compared to the Fourier-based prior, as measured by the auPRC of importance overlap with ChIP-seq peaks.

Figure S26: Specific examples of attributions under various levels of GC content. For each level of GC bias, we select a sampled input sequence and examine the DeepSHAP attributions. GC content ranges from +0% to +4%: a GC bias of $+x\%$ means the probability of G or C in the background of a positive sequence is $(50 + x)\%$, while the background of a negative sequence has a G/C probability of 50% (i.e. no bias at all). We display the value of the DeepSHAP importance for the bases present along the entire input sequence, as well as a zoomed-in view of the region surrounding the central motif. We also show the "hypothetical" attributions along each input sequence (i.e. the importance that would be given to each of the four bases at each position, even if that base were not in the input sequence—see Supplementary Methods Sec. 3).