[Reviews · NeurIPS 2020]

Review 1

Summary and Contributions: The authors introduce a novel regularization technique based on Fourier transform of input gradients that encourages neural network models to emphasize contiguous segments in the input data. The authors developed this technique for predicting transcription factor binding from DNA sequences and seek to annotate functional motifs in the DNA. For such tasks, the authors demonstrate superior performance and stability with their regularizer.

Strengths: The Fourier regularization technique is clever and may have applicability for other ML problems, or at least inspire further work on training interpretable models. The authors use many different views and orthogonal data to argue for the benefit of their regularizer.

Weaknesses: The most cited methods in this space use a different pooling architecture and train multi-task for dozens of epochs. In contrast, the authors train single task and choose the model achieved after one or two epochs as best. These differences may contribute to the rapid overfitting and saliency variance. Do multi-task models trained for longer improve motif annotation? Does your method also improve motif annotation in that framework? Multi-task datasets can be obtained from both the Basset and DeepSea papers. Does the regularizer improve training using their model architectures? It’s argued that the method is generally applicable. However, it requires binary peak annotations because only positive examples apply the Fourier regularization. Does motif annotation suffer if the regularization is applied to all sequences? If so and if that is not recommended, then the authors should clarify that binary peaks are required. Why is a smoothing operation added before the Fourier operation? Does performance suffer without this? Several questions about Figure 2. First, how do you divide Fourier components into low and high frequency? Second, how do you compute entropy of the nucleotide attributions, which don't represent a probability distribution? Third, in panel C, the zoomed in range does not fit within the ostensibly larger range in the top browser view. In cases where the motif instance does not match the optimal kmer, some nucleotide(s) could be mutated to increase the prediction and you’d observe a negative attribution score sandwiched in between positive attribution scores. This regularization wouldn't seem to be as appropriate then. You might take a look at some examples like that and consider adding a supplementary figure to explore and discuss this scenario. ———————————————— I have read the other reviews, and the authors’ response. To my first point, the authors point out that many of their experiments are multi-task training data sets. Reviewing section 2.2, I see that’s true, but the multiple tasks always represent the same TF. This is quite different from the “massive multi-task“ framework employed by the most cited methods in the space. The authors declined to address their use of different architectures. The authors hypothesize that their models also overfit rapidly because the datasets are large, for example 700k examples for the DNase K562. I find this to be backwards; as the number of sequences in the data set grows, the influence of each single sequence on the model parameters decreases. Overfitting on the second epoch is strange. However, the authors have a lot of experience working with these models and data. If this is normal for them, then hopefully the comparisons are valid and generalizable. To my second point, the authors committed to clarifying this point in the text. To my third and fifth points, the authors clarified their method. They declined to perform an experiment without the smoothing operation, but their response helped me understand the value that it brings. To my fourth point, the authors clarified that they normalize the attributions to represent pseudo-probabilities. Based on this discussion, I am willing to increase my score for the paper to 5 and support its acceptance.

Correctness: Yes.

Clarity: Yes.

Relation to Prior Work: Yes.

Reproducibility: Yes

Additional Feedback:


Review 2

Summary and Contributions: The authors propose a novel Fourier-based attribution prior to improve the interpretability of deep learning models of transcription. The methods show promise relative to the baseline, with improvements almost uniformly across the board in both the predictive accuracy on a test set as well as stability of the derived motifs.

Strengths: 1) The Fourier prior is well-motivated, and the resulting interpretability is also compared to an independent method, DeepSHAP, based on Shapely values. 2) The results show robustness of the predictions and the method's ability to recover known transcription factors such as CLOCK. 3) The authors analyze multiple facets of their methodology on the same 4 datasets, demonstrating an improvement on almost all of them.

Weaknesses: 1) It is unclear how much better these methods perform than the "old-school" motif prediction methods. 2) The only comparison is between models with the prior and models without the prior, there are no other methods being compared to. 3) The authors do not seem to have a good explanation for the situations in which their method performs less well than the baseline.

Correctness: The methodology appears to be correct to the extent that I am able to ascertain it.

Clarity: The paper is clearly written and coherently argued.

Relation to Prior Work: Only similar deep learning models are discussed; other interpretable, but not "deep", models are not discussed at all.

Reproducibility: Yes

Additional Feedback:


Review 3

Summary and Contributions: The authors propose a novel attribution prior for training deep learning models in genomics. At training time they obtain attribution scores through input gradients and regularize the model by penalizing high-frequency components of their Fourier components. Throughout extensive empirical experiments, the authors show that the proposed attribution prior dramatically improve models’ stability, interpretability, and generalization performance.

Strengths: (1) Novelty and Significance The proposed Fourier-based attribution prior is novel and effective. It stems from the intuitive motivation that attribution scores should be focused on low-frequency reasonable motifs. To the best of my knowledge, this is the first work to propose Fourier-based priors and also one of the first works to use attribution priors to regularize deep learning models in genomics. The extensive empirical experiments are impressive and show that it would benefit wide NeurIPS and Bioinformatics community. (2) Reproducibility While the main manuscript focuses on presenting the quantitative and qualitative experiment results, the authors also provide detailed information and codes to reproduce the experiments through supplementary materials. I highly appreciate the authors’ effort in providing all the information to reproduce the results.

Weaknesses: I did not find any major limitations of the work, but I would also like to see some more experiments and authors' opinions on the following issues. (1) There is a discrepancy of attribution methods used during training and evaluation. Considering that different attribution methods often produce different scores, why would penalizing input gradients improve the SHAP scores? Does the proposed method also improve the interpretability results in terms of the input gradients? How much different attribution methods (used either training or evaluation) would affect the proposed method? (2) Although there are some previous works that use attribution priors to regularize deep learning models, the manuscript does not cover them. Please include related works in the manuscript. (3) While the authors qualitatively show that the attribution priors improve the detection of motifs, can you provide quantitative results compared to ground-truth motifs? I am not sure, but the TomTom tool might help comparing obtained motifs with ground-truth motifs.

Correctness: The claims and methods seem correct, but I would like to recommend more ablation studies on the encoding scheme.

Clarity: The paper is generally well written and easy to understand. The contribution is clear, and the way the authors claim is well understandable and notations are explained well.

Relation to Prior Work: No. I think the authors should properly cite and explain related works that use attribution priors to regularize deep learning models.

Reproducibility: Yes

Additional Feedback: No additional comments. ---- Post Author Feedback Comments --- I have read the Author Feedback and I'm happy that the authors have clarified some points.


Review 4

Summary and Contributions: The stated contribution of the paper is an attribution prior penalizing high-frequency components of Fourier spectrum of per-position attribution scores that the authors claim improves interpretability and stability of these models. The motivation of the paper is driven by challenges encountered from applying deep learning models to nucleotide sequences to predict measurements from functional genomics (i.e. transcription factor binding and chromatin accessibility). These models have improved prediction performance but are difficult to interpret, leading to attribution scores as one approach to interrogate the model and inputs for sequence motifs (and coordination between motifs) that drive individual predictions. However, the challenge is that attribution scores can be noisy and sensitive to random initializations of the same model. The manuscript empirically demonstrates that the addition of their per-position attribution prior to the training of their models of transcription factor binding (binary outcomes or TF profiles) improves not only detection of motifs within individual models but stability of detection of these motifs between random initializations of the model.

Strengths: The authors nicely demonstrate improvements in detection and stability of motifs using their attribution prior. Quantifying improvements in attribution can be challenging, but the authors provide several metrics to this point and compelling visual examples. The empirical analysis is thorough. Using the frequency spectrum of the attributions as a proxy for localization/noisiness for penalization is a nice contribution to bring in to this area from other fields that consider high frequency signals as noise The manuscript demonstrates that their attribution prior does not decrease performance on the original predictions of interest

Weaknesses: 1. The paper did not compare to related works (citations [9] and [10]) attribution priors, only added L2 regularization * In particular, [9] describes an image-classification prior which penalizes total variation between adjacent pixels. This implicitly penalizes high-frequency modes of the attribution vector (image in this case) and is closely related to total variation image denoising (vs the manuscript’s method’s relationship to low-pass filters). A comparison to some of the priors in these previous works (adapted to 1D sequences) would have been appropriate. * The impact of other forms of regularization (dropout, L1, etc.) would have also been nice to see 2. It would have been helpful to directly clarify in the description of the methods if/when other forms of regularization (L2, L1, dropout) were being used and/or specifically if there was no regularization being used to prevent overfitting by default (No prior model in tables 1-4). Since the paper claims in Tables 1-4 that in addition to improved interpretability, the fourier transform attribution prior offers performance improvement, it becomes important to specify if the model with the Fourier transform prior has better/similar/worse results when compared to the default model with some sort of commonly used regularization (L2, L1, dropout). They should show that the performance doesn’t degrade when their prior is compared to a default model with these other regularizations (L2, L1, dropout). 3. The supplemental code was missing configuration files or driver scripts to indicate which parameters/models were used for which figures. The train model scripts had default parameters specified using the Sacred library/framework, and they and several of the notebooks were dependent on hardcoded paths and files that do not exist in the code (or as targets for downloading by other scripts, at least as far as we could assess). 4. Details about what was being computed and when were sometimes sparse. For example, the paper quantifies differences in Shannon entropy of the attributions. How is this being computed (i.e. what is the probability distribution being used/inferred)? 5. The attribution method g must be differentiable almost everywhere in order to permit optimization of the attribution loss using gradient descent. This limitation should be clearly stated although all evaluations in the paper used a input feature-scaled gradient for g. 6. The manuscript sets a frequency threshold for penalization in relationship to a motif length of 7. This is specific to the TF prediction task. Other predictions from nucleotide sequences (i.e. RNA-binding protein predictions, etc.) may involve sequence motifs that are traditionally considered to be more degenerate and short. We suspect that the approach might have difficulty penalizing high frequency components without impacting predictions, and we suggest that the potential limitations for detecting real but short regulatory regions be discussed when considering its broader impact. 7. Tables 3 and 4 show worse performance in auPRC for Nanog/Oct4/Sox2. * It is unclear why the better performance for the “no prior” model was not bolded. * The setting of “more complex motif sequence syntax” is a reason DL in genomics is particularly interesting and it was disappointing to see lower performance in this setting. While the manuscript explains that other motifs that were not underlying a peak were also being highlighted, referring to the example in S20, further quantification of this phenomenon and additional discussion of how it could cause lower performance in this evaluation is warranted. ====== We have read the authors response and the other reviews. We thank the authors for their detailed response, especially with respect to our questions/comments about other choices for regularization. We felt that the original manuscript implied that the Fourier regularization penalty improved interpretability without decreasing predictive performance. However, the response indicates that the penalty decreases predictive performance compared to traditional regularization. The authors should explicitly note that it has worse predictive performance in these cases as part of ensuring they do not overstate their results. We take issue with the reasoning for leaving out a comparison between traditionally-regularized models with/without additional Fourier regularization penalty. Regularization of several competing penalties is very common in the literature. Difficulties in combining this penalty with other forms of regularization is also an observation that should be noted. Ultimately, these issues do not impact our overall score because the results on interpretability are the focus and still compelling, but we hope that the authors are more forthcoming about their method’s limitations in the final manuscript.

Correctness: We did not identify any errors and were satisfied with their empirical approach.

Clarity: Yes, the paper is well written.

Relation to Prior Work: While the manuscript cites two papers ([9], [10]) describing attribution priors, they are not discussed nor used as a standard for comparison. In particular, [9] describes a total variation attribution prior on adjacent pixels (i.e. positions) that would implicitly penalize high-frequency components of the attribution vector’s Fourier spectrum.

Reproducibility: No

Additional Feedback: 1. The work was reasonably reproducible, but there were issues with the supplemental code that were discussed in the weaknesses section. 2. “Dramatically” improves in abstract is uninformative, if not also overstated. In general, while the authors show consistent improvements using various metrics, in many cases those appear small and/or not statistically significant. Thus, toning down the associated claims/statements seems to be warranted. 3. Implicit periodicity of the Fourier transform over the length of the input window could potentially lead to weird artifacts in penalties when attribution signal is present at both ends of the attribution vector (which Fourier transform would consider adjacent)

[Author Response · NeurIPS 2020]

**Response to R1**: Thank you for catching the typo in Fig 2C. Re. **suboptimal k-mers**: our prior regularizes the *absolute*
*value* of the gradients (Supp. Methods 2.3), and these absolute gradients are smoothed, thus it can handle negative
or low-scoring bases within a motif (this is also why the **smoothing** is important). Details on dividing the Fourier
spectrum into **high vs low frequencies** are in Sec. 2.1 of the main text and 2.3 of the Supp. Methods. We will clarify
the need to designate a subset of regions to which the prior is applied. Re. **multi-tasked models**: the vast majority of
our models are trained on multiple tasks (and our prior improves interpretability on them, as well). We used our binary
model formulation because we found that Basset's style of massively multi-tasked models achieved worse performance
(e.g. fine-tuning a massively multi-tasked Basset model on the binary K562 accessibility task gets a test-set auROC of
0.953 and auPRC of 0.502, and the performance is worse without fine-tuning; in contrast, our architecture trained from
scratch gets an auROC of 0.966 and auPRC of 0.537). This is also why our models train in **fewer epochs**: due to the
reduced number of tasks, the models converge to a good optimum quicker. Note that our epochs are large (e.g. for K562
accessibility, we have 714,423 positive and negative examples per epoch; for more details, see Supp. Methods 2.1).

**How does the Fourier-based prior compare to other attribution prior formulations [R2,3,4] and traditional**
**regularization [R4]?** We train binary SPI1 models with the smoothness prior [Erion et al., 2019] over several random
initializations, picking the prior weight identically to how we picked the Fourier-based prior's weight (i.e. so that the
prior loss is on a similar scale as the correctness loss, as recommended by Ross et al. [2017]). We found that the prior
*severely* degrades predictive performance (validation set prediction loss was 0.279 w/ smoothness prior, 0.269 w/ no
prior, and 0.262 w/ Fourier-based prior), while only *marginally* improving importance overlap with peaks (auPRC of
0.595 w/ no prior, 0.616 w/ Fourier-based prior, and 0.618 w/ smoothness prior). This is consistent with the intuition
that the smoothness prior penalizes *all* transitions in attributions, thereby encouraging the model to find fewer motifs
and to give them less importance. *In fact, the smoothness prior has worse predictive performance than having no prior*
*at all* ($p < 1 \times 10^{-4}$ by Mann–Whitney U-test). In comparison, the Fourier-based prior does not penalize motifs at all,
provided the transitions are somewhat smooth. We then tune the smoothness prior weight and pick the model with the
lowest validation loss. With a smoothness prior weight that is orders-of-magnitude lower, we somewhat rescue the
predictive performance (validation loss 0.268, whereas the Fourier-based prior still gets a better validation loss of 0.262),
but the smoothness prior also loses its edge in interpretability over the Fourier-based prior (smoothness prior auPRC of
peak overlap drops to 0.601). We perform this same comparison with the "sparsity" prior defined in Erion et al. [2019],
in which models are rewarded for sparse attributions. Compared to the Fourier-based prior, the sparsity prior shows
the same trends as the smoothness prior (after tuning the sparsity prior loss weight, the sparsity prior achieves a peak
overlap auPRC of 0.597, which is almost as bad as no prior; and a validation loss of 0.269, still slightly worse than
the Fourier-based prior). Of these three different prior formulations, only the Fourier-based prior is able to maximize
both interpretability *and* predictive performance. **Re. comparison to traditional regularization:** On our SPI1 binary
models, L2-regularization alone did improve predictive performance more than the Fourier-based prior. *However*, this
was at the severe cost of interpretability (Sec. 8 of the main text & Supp. Fig. S23–S24). In fact, *L2-regularization*
*hurts interpretability worse than no prior/regularization at all* (peak overlap auPRC with L2 is 0.585, versus 0.595
without any prior). We found that training models with *both* the prior *and* L1/L2-regularization was challenging, as it
requires simultaneous optimization of 3 competing losses. Thus, to be consistent in our comparisons, we trained our "no
prior" models without L1/L2-regularization. Note that profile models aren't typically trained with L1/L2/dropout, per
Avsec et al. [2019]. Harmoniously merging our prior with traditional regularization is a good direction for future work.

**Why did the auPRC of peak overlap not improve in some cases with the Fourier-based prior? [R2,4]** The
Fourier-based prior's improvements were consistent and *statistically significant* in the vast majority of experiments (as
shown in the manuscript). The only time it underperformed was on the precision–recall of importance overlapping
peaks/footprints, and *only* on complex tasks with *binary* models (on profile models, the prior always improved the
auPRC). We emphasize that this is not a failure of the prior, but a symptom intrinsic to the binary architecture. *Profile*
models are able to finely track motifs along shifting peaks, so they can isolate the specific motifs underlying peaks.
*Binary* models, however, see the same sequences repeatedly to predict a binary label, so if a sequence has multiple
motifs (i.e. for complex tasks like predicting Nanog/Oct4/Sox2 binding or chromatin state), the model would also rely
on motifs that do not underlie peaks/footprints. The Fourier-based prior improves the identification of *all* relevant motifs
and increases their relative importance. Thus, on a binary architecture, the prior highlights motifs outside of ChIP-nexus
peaks or DNAse-seq footprints (as shown in the manuscript), because they are *still informative for a binary label*.
On this intrinsically limited architecture, this is the correct thing to do, even though it directly reduces the computed
precision of importance overlap in peaks/footprints for complex tasks.

**Why does penalizing gradients improve DeepSHAP scores? [R3]** Gradients can serve as a first-order approximation
of other measures of importance and are easily implemented in popular frameworks. We showed attributions with
DeepSHAP precisely to demonstrate that our prior benefits interpretability across *multiple* different measures of
importance (in fact, the improvements are even more striking in gradient space).

**Shannon entropy? [R1,4]** We normalize the magnitude of the attributions along the sequence into "probabilities".

[Meta-Review · NeurIPS 2020]

This paper proposes a novel Fourier-based attribution prior, which can facilitate the application of deep learning to sequence data by improving the interpretation of the learned model. There is significant technical novelty, the authors presented strong experimental results, and the proposed approach is likely of high impact in the field of genomics. Reviewers are largely satisfied with the author feedback. Therefore, the submission is clearly above the bar for the acceptance to NeurIPS.